# Mixture of Link Predictors on Graphs

**Li Ma**[1*†‡] , **Haoyu Han**[2*] , **Juanhui Li**[2] , **Harry Shomer**[2] , **Hui Liu**[2] ,
**Xiaofeng Gao**[1†] , **Jiliang Tang**[2]
[1]Shanghai Jiao Tong University, [2]Michigan State University
`mali-cs@sjtu.edu.cn, gao-xf@cs.sjtu.edu.cn`
`{hanhaoy1,lijuanh1,shomerha,liuhui7,tangjili}@msu.edu`

## Abstract

Link prediction, which aims to forecast unseen connections in graphs, is a fundamental task in graph machine learning. Heuristic methods, leveraging a range of different pairwise measures such as common neighbors and shortest paths, often rival the performance of vanilla Graph Neural Networks (GNNs). Therefore, recent advancements in GNNs for link prediction (GNN4LP) have primarily focused on integrating one or a few types of pairwise information. In this work, we reveal that different node pairs within the same dataset necessitate varied pairwise information for accurate prediction and models that only apply the same pairwise information uniformly could achieve suboptimal performance. As a result, we propose a simple mixture of experts model Link-MoE for link prediction. Link-MoE utilizes various GNNs as experts and strategically selects the appropriate expert for each node pair based on various types of pairwise information. Experimental results across diverse real-world datasets demonstrate substantial performance improvement from Link-MoE. Notably, Link-MoE achieves a relative improvement of 18.71% on the MRR metric for the Pubmed dataset and 9.59% on the Hits@100 metric for the ogbl-ppa dataset, compared to the best baselines. The code is available at `https://github.com/ml-ml/Link-MoE/`.

## 1 Introduction

Link prediction (LP) is a central challenge in graph analysis with many real-world applications, such as recommender systems [1, 2], drug discovery [3], and knowledge graph completion [4, 5]. Specifically, LP attempts to predict unseen edges in a graph. Unlike node-level tasks where Graph Neural Networks (GNNs) excel in modeling *individual* node representations [6], LP demands the use of *pairwise* node representations to model the existence of a link where vanilla GNNs often fall short [7, 8]. Traditionally, various heuristic methods [9] were used to identify new links by encapsulating the pairwise relationship between two nodes. For instance, the Common Neighbors (CN) heuristic [10] counts the number of shared neighbors between a node pair, postulating that the number of common neighbors can indicate the likelihood of a connection. The Katz index [11] considers the total number of paths between two nodes, assigning a higher weight to those shorter in length. Node feature similarity-based methods [12] assumes that nodes with similar features tend to connect. Despite their simplicity, these heuristic-based methods are still considered as strong baselines in LP tasks.

To leverage both the representational power of GNNs and the effectiveness of heuristics, recent GNN4LP works have sought to incorporate pairwise information into GNN frameworks, thereby

---

[*]Equal Contribution.

[†]Li Ma and Xiaofeng Gao are in the MoE Key Lab of Artificial Intelligence, Department of Computer Science and Engineering, Shanghai Jiao Tong University.

[‡]This work is done when Li Ma is a visiting student at Michigan State University.

enhancing their expressiveness for better link prediction. For example, NCN/NCNC [13] exploit the common neighbor information into GNNs. Neo-GNN [14] further incorporates multi-hop neighbor overlap information. SEAL [15] and NBFNet [16] leverage the full and partial labeling trick, respectively, to indict the target node pair, which has been proven to learn heuristic patterns like common neighbors and the Katz index. These GNN4LP methods mark significant progress in link prediction [17].

However, both traditional heuristic approaches and GNN4LP models typically adopt a one-size-fits-all solution, uniformly applying the same strategy to all target node pairs. There are several limitations with this one-size-fits-all solution: **(1) Limited Use of Heuristics:** These methods utilize only one or a few heuristics. Our preliminary studies in Section 3 have shown that different heuristics tend to complement each other, and employing multiple heuristics within the same dataset can lead to improved link prediction performance. Therefore, methods relying on a single type of heuristic might not be optimal. **(2) Uniform Application of Heuristics**: In Section 3, we also find that different node pairs within the same dataset often require distinct heuristics for predictions. Consequently, these methods that uniformly apply the same heuristic across all node pairs lack this adaptability, potentially leading to suboptimal performance. These findings underscore the pressing need for an approach that can adaptively apply a range of pairwise information specific to node pairs. Inspired by the superior performance of existing GNN4LP, we delved deeper into understanding different GNN4LP models. Our investigation reveals that these models are highly complementary and they excel under specific conditions, often correlated with particular heuristics. For example, the NCN model tends to perform well in scenarios with a high number of common neighbors. These observations motivate us to ask: *can we design a strategy that can simultaneously enjoy the strengths of various GNN4LP models and correspondingly enhance link prediction?*

In response to this question, we introduce a simple yet remarkably effective mixture of experts model for link prediction – **Link-MoE**. This model operates by utilizing a range of existing link predictors as experts. A gating function is learned to assign different node pairs to different experts based on various types of pairwise information. Extensive experimental results showcase the surprisingly effective performance of Link-MoE. For instance, it surpasses the best baseline on Pubmed and ogbl-ppa dataset by 18.71% on the MRR and 9.59% on the Hits@100, respectively.

## 2 Related Work

### 2.1 Link Prediction

Link prediction aims to predict unseen links in a graph. There are mainly three classes of methods for the link prediction task. An overview of each is given below.

**Heuristic Methods**: Heuristic methods have been traditionally used for link prediction. They attempt to explicitly model the pairwise information between a node pair via hand-crafted measures. Several classes of heuristics exist for link prediction [17] including: local structural proximity, global structural proximity, and feature proximity. *Local Structural Proximity* (LSP): These method extract the information in the local neighborhood of a node pair. Common Neighbors (CN) [10], Adamic-Adar (AA) [18], and Resource Allocation [19] are popular measures that consider the number of shared 1-hop neighbors between the node pair. *Global Structural Proximity* (GSP): These methods attempt to model the interaction of a node pair by extracting the global graph information. The Shortest Path Distance assumes that a shorter distance between nodes results in a higher likelihood of them connecting. Both Katz Index [11] and Personalized Pagerank (PPR) [20] consider the paths of disparate length that connect both nodes, giving a higher weight to those shorter paths. *Feature Proximity* (FP): FP measures the similarity of the node features for both nodes in the pair, positing that nodes with similar features are more likely to form edges. Previous work [21, 22] has derived heuristics algorithms to measure the feature proximity.

**GNN-based Methods**: Recent work has looked to move beyond pre-defined heuristic measures and model link prediction through the use of graph neural networks (GNNs). Earlier work [23, 24, 25] has sought to first use a GNN to learn node representations, and the representations for both nodes in a pair are then used to predict whether they link. However, multiple works [7, 8] have shown that node-based representations are unable to properly model link prediction. As such, newer methods, which we refer to as GNN4LP, attempt to learn pairwise representations to facilitate link prediction. Both SEAL [7] and NBFNet [16] condition the GNN aggregation on either both or one of the two

nodes in the pair. While expressive, both methods tend to be prohibitively expensive as they require aggregating messages separately for each node pair. Recent methods [14, 13, 26, 27] have attempts to devise more efficient ways of learning pairwise representations by injecting pairwise information in the score function, bypassing the need of customizing the GNN aggregation to each node pair. These methods typically attempt to exploit different structural patterns on the local and global scales.

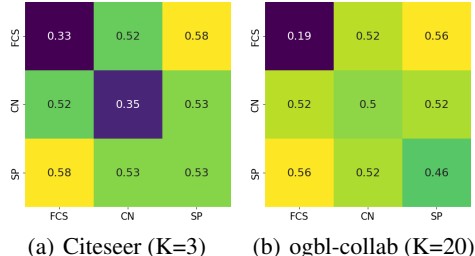

(a) Citeseer (K=3)     (b) ogbl-collab (K=20)

Figure 1: Hits@K of combined heuristics.

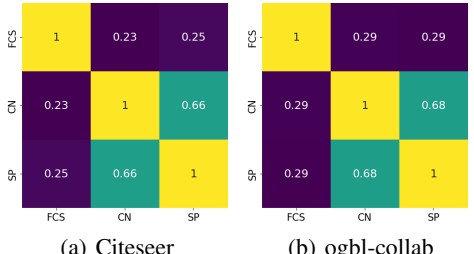

(a) Citeseer          (b) ogbl-collab

Figure 2: The overlapping ratio of heuristics.

**Ensemble Methods**: Ensemble-based methods for link prediction primarily fall into two categories: bagging and stacking. Bagging methods create multiple base learners trained on varied data subsets and integrate their predictions for final output [28, 29]. Stacking methods train multiple models on the entire graph and use a meta-model to integrate their predictions ( [30, 31, 32, 33]). However, these methods directly combine the predictions of base learner without considering the specific patterns and heuristics of base learners.

## 2.2 Mixture of Experts

The use of Mixture of Experts (MoE) [34, 35], which is based on the divide-and-conquer principle to divide problem to different experts, has been explored across various domains [36]. Recent researches mainly focus on the efficiency of leveraging the MoE in the NLP [37, 38] or Computer Vision [39] domains. Chen et al. [40] attribute the success of MoE to the cluster structure of the underlying problem. In the graph domain, GMoE [41] integrates the MoE model with GNNs, enabling nodes to learn from information across different hops. Wu et al. [42] leverage MoE to address the distribution shift issue in GNNs. To the best of our knowledge, Link-MoE represents the first instance of MoE specifically tailored for the link prediction task, showcasing exceptionally effective performance.

## 3 Preliminary

In this section, we begin by analyzing the relationship between various heuristics employed in link prediction, aiming to uncover the complex patterns that exist within the same dataset. Subsequently, we explore the relationship between the performance of different GNN4LP models and these heuristics. This exploration aims to identify the most suitable scenarios for each model, thereby enhancing our understanding of how different approaches can be optimally applied in varying link prediction contexts. Before that, we first introduce key notations and experimental settings.

**Notations and experimental settings.** Let $\mathcal{G} = (\mathcal{V}, \mathcal{E})$ be a graph with $n$ nodes, where $\mathcal{V}$ is the node set and $\mathcal{E}$ is the edge set. $\mathcal{N}_i$ denotes the neighborhood node set for node $v_i$. The graph can be denoted as an adjacency matrix $\mathbf{A} \in \mathbb{R}^{n \times n}$, and each node $v_i$ may be associated with a $d$-dimensional feature $\mathbf{x}_i$ and we use $\mathbf{X} = [\mathbf{x}_1, \ldots, \mathbf{x}_n]^\top \in \mathbb{R}^{n \times d}$ to denote the node feature matrix. We conduct analysis on the Cora, Citeseer, Pubmed, ogbl-collab, and ogbl-ppa datasets. We use Hits@K as a metric to measure the ratio of positive samples ranked among the top $K$ against a set of negative samples. The details on each dataset and the evaluation setting can be found in Appendix A. Due to the limited space, we only illustrate partial results in the following subsections. More results can be found in Appendix B.

### 3.1 Exploring Heuristics in Link Prediction

In this subsection, we focus on three vital types of pairwise factors used in link prediction identified by Mao et al. [17]: (a) local structure proximity, (b) global structure proximity, and (c) feature

proximity. For each type, we adopt a single widely used heuristic as a representative metric including Common Neighbors (CN) [10] for local structure proximity, Shortest Path (SP) [43] for global structural proximity, and Feature Cosine Similarity (FCS) for feature proximity. We initially evaluate the performance of each heuristic individually and then assess their combinations by simply adding their normalized values. Specifically, we normalize each heuristic value ($h$) to the range of [0, 1] using $\frac{h - h_{max}}{h_{max} - h_{min}}$, where $h_{max}$ and $h_{min}$ are the maximum and minimum heuristic values in the dataset. One exception is the calculation of SP, where a smaller SP indicates a higher likelihood that two nodes are connected. Therefore, we first calculate $\frac{1}{SP}$, and then normalize it in the same way as other heuristics. For this evaluation, we employ Hits@3 as the metric for smaller datasets and Hits@20 for larger OGB datasets. The results for Citeseer and ogbl-collab are illustrated in Figure 1, with diagonal values representing the individual performance of each heuristic. We can have two observations: (**1**) combining different heuristics generally enhances overall performance, indicating that reliance on a single heuristic may be inadequate for accurate link prediction; and (**2**) the performance of each heuristic varies across datasets. For instance, in the Citeseer dataset, FCS and CN exhibit comparable performance. However, in the ogbl-collab dataset, CN significantly outperforms FCS.

We further investigate the overlap in correctly predicted sample pairs by each heuristic. We use the Jaccard Coefficient to calculate the overlapping ratio between each pair of heuristics. For the calculation of the Jaccard coefficient, we use the Hits@K metric for each edge. Specifically, we choose Hits@3 for small datasets and Hits@20 for the OGB datasets. We first rank the prediction scores of each method for both positive and negative edges. If the prediction score of a positive edge is in the Top-K, we label this positive edge as 'present' and add it to the correct prediction set. In this way, we can calculate the Jaccard coefficient by comparing the correct prediction sets for each pair of methods. The results for the Citeseer and ogbl-collab datasets are presented in Figure 2. We observe that the overlapping ratio between certain heuristics is notably low. This implies that the sets of node pairs correctly predicted by different heuristics have a minimal intersection. Therefore, **different node pairs require distinct heuristics for accurate prediction even on the same dataset**.

From the previous analysis, it is enticing to think that simply considering multiple heuristics should result in superior link prediction performance. We test this hypothesis by learning to classify links using multiple popular heuristic methods. For a single link, the individual heuristic scores are concatenated together and passed to an MLP, where the output is then used to classify the link. The full set of heuristic considered can be found in Section 5.1. The results on Citeseer and ogbl-collab can be found in Table 1. We report the MRR for Citeseer and Hits@50 for ogbl-collab. We find that ensembling multiple heuristics can modestly improve the performance. However, it still noticeably lags behind GNN4LP methods in performance. Based on this observation, we are motivated to investigate whether different GNN4LP models can be used to model a wider variety of links.

|  | Method | Citeseer | ogbl-collab |
|---|---|---|---|
| Heuristic | CN | 28.34 | 61.37 |
|  | Shortest Path | 31.82 | 46.49 |
|  | Katz | 38.16 | 64.33 |
|  | Feature Similarity | 31.82 | 26.27 |
|  | **Ensemble** | 44.08 ± 0.18 | 64.44 ± 0.21 |
| GNN4LP | Neo-GNN | 53.97 ± 5.88 | 66.13 ± 0.61 |
|  | NCNC | 64.03 ± 3.67 | 65.97 ± 1.03 |

Table 1: Performance of ensembling heuristics.

## 3.2 Exploring GNN4LP Models and Heuristics

In this subsection, we move beyond analyzing the performance of only heuristic measures and further consider the capabilities of different GNN4LP methods. To better understand the abilities of different GNN4LP models, we first evaluate the overlapping ratio between different models using the Jaccard Coefficient. We also include the MLP and different heuristics in the analysis. The overlapping ratio of different methods on ogbl-collab dataset is shown in Figure 3. These results reveal that the overlapping ratios among different GNN4LP models are rel-

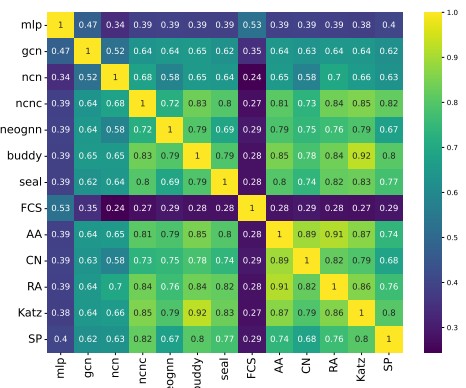

Figure 3: The overlapping ratio on ogbl-collab.

atively low, suggesting that each model is capable of predicting a unique set of links. Furthermore, different GNN4LP models have varying degrees of overlap with different heuristics. These observations lead us to an intriguing question: *Are the unique sets of links correctly predicted by different GNN4LP models related to specific heuristics?*

To answer this question, we first categorize node pairs into 5 groups based on each heuristic and evaluate the performance of different GNN4LP models within these groups. Additionally, we also include the MLP and GCN in our analysis. The performance of these models across different Common Neighbors (CN) groups for Cora and ogbl-collab is depicted in Figure 4. The x-axis represents different groups, along with the proportion of node pairs in each group. From the results, we can find that **no single model consistently outperforms others across all groups** on either dataset. Interestingly, when there are no common neighbors, MLP and GCN tend to excel in both the Cora and ogbl-collab datasets. In situations with a few common neighbors, SEAL shows better performance in the Cora dataset, while BUDDY tends to lead in the ogbl-collab dataset. With an increase in the number of common neighbors, methods that encode CN information, such as NCNC, generally exhibit strong performance. A similar phenomena can be found on other datasets and heuristics in Appendix B.

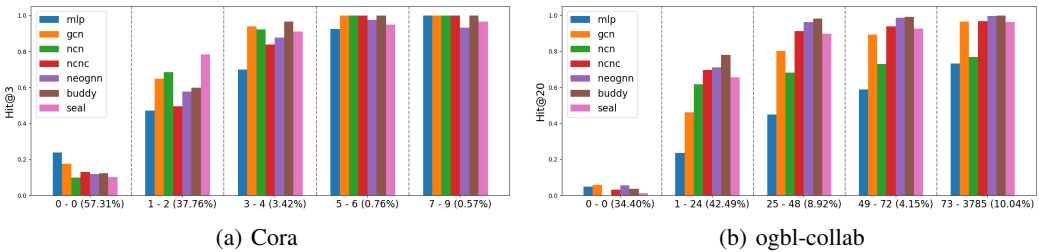

(a) Cora          (b) ogbl-collab

Figure 4: The performance of different models on each CN group.

In conclusion, our analysis underscores that accurate link prediction necessitates the use of multiple heuristics. Different node pairs require distinct heuristics for optimal prediction. Furthermore, the overlapping ratio of different GNN4LP models is relatively low. The performance of powerful GNN4LP models is closely tied to the specific heuristics they encode. And there is no single model that can uniformly achieve the best performance across all scenarios. These findings pave us a way to adaptively select the most suitable model for each node pair to achieve better overall performance in link prediction task.

## 4 Method

The investigations conducted in Section 3 reveal that various heuristics complement each other for the task on link prediction. This implies that different node pairs may need different heuristics to properly predict the existence of a link. Therefore, one approach to potentially achieve better performance in link prediction is to integrate all these heuristics into a single, unified model. However, as shown in Table 1, this strategy fails to outperform existing GNN4LP methods. This suggests that these GNN4LP models are already quite effective. Our findings further suggest that different GNN4LP models demonstrate unique strengths in different scenarios, which are related to specific heuristics. Therefore, leveraging these diverse heuristics as a guidance could help identify the most suitable GNN4LP models for specific node pairs. Based on the above intuitions, we aim to design a framework that can harness the unique strengths of each model to enhance the performance of link prediction.

### 4.1 Link-MoE – A General Framework

To leverage the strengths of various existing models, we introduce a novel mixture-of-experts (MoE) method tailored for link prediction, which we term – **Link-MoE**. An overview of Link-MoE is depicted in Figure 5(a). There are two major components: the gating model and the multiple expert models. The gating function can be implemented using any neural network and each expert can be any method used for link prediction. When predicting whether a node pair $(i, j)$ are linked, the gating function utilizes their heuristic information to produce normalized weights for each expert.

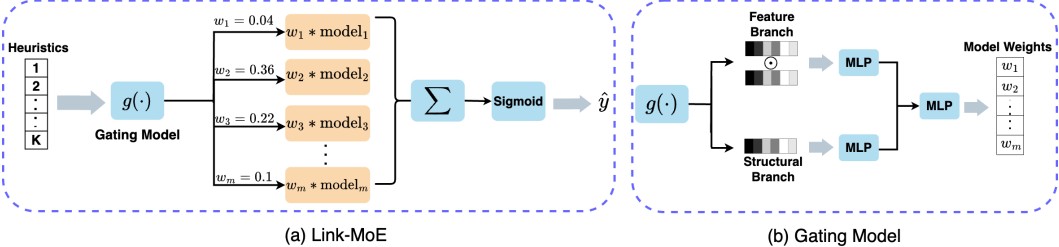

Figure 5: An overview of the proposed Link-MoE.

These weights dictate the level of contribution each expert model has towards the final prediction. Each expert model processes the graph and node pair information, estimating the likelihood of a connection between the two nodes. The individual expert predictions are then aggregated according to the weights assigned by the gating function. The final prediction is made using the sum of the weighted scores, effectively leveraging the strengths of multiple experts to determine the probability of a link. Formally, the prediction of Link-MoE can be expressed as:

$$Y_{ij} = \sigma \left( \sum_{o=1}^{m} G(\mathbf{x}_{ij}, \mathbf{s}_{ij})_o E_o(\mathbf{A}, \mathbf{X})_{ij} \right), \tag{1}$$

where $m$ denotes the number of expert models incorporated, pairwise node features $\mathbf{x}_{ij}$ and structural heuristics $\mathbf{s}_{ij}$ serves as the input to the gating function, $G(\cdot)$ represents the gating model function, $E_o$ refers to the $o$-th expert model, and $\sigma$ is a sigmoid activation function. This configuration makes the Link-MoE remarkably flexible and easily adaptable, allowing for the seamless integration of different expert models as required. We will detail our implementation in the following subsections.

## 4.2 The Design of the Gating Model

As highlighted in our preliminary studies (Section 3), it's evident that different GNN4LP models excel in varying contexts and that their strengths can be indicated by different heuristics. Therefore, we incorporate a broad spectrum of heuristics as inputs to the gating model to leverage these strengths. Specifically we consider CN [10], AA [18] and RA [19] to model the local structural proximity and Shortest Path, Katz index [11], and PPR [44] for the global structural proximity. For the feature proximity, to ensure the score is invariant to the ordering of the node pair, we utilize the element-wise product for deriving the feature heuristic of node pair $(i, j)$. This is defined as $\mathbf{x}_{ij} = \mathbf{x}_i \odot \mathbf{x}_j$. We also use $\mathbf{s}_{ij} = [s_{ij}^1, s_{ij}^2, ..., s_{ij}^k]$ to represent all the $k$ structural heuristics, and the dimension of each structural heuristics is typically very small. However, the node pair feature heuristic is equal to the input node feature dimension and can span hundreds or even thousands of dimensions. This significant disparity in dimensionality could hinder the model's ability to effectively learn from the structural heuristics. To address this challenge, we design a two-branch gating model, as illustrated in Figure 5(b). Each branch is a simple MLP. One branch is dedicated to encoding the structural heuristics, while the other focuses on processing the node pair features. After that, these two branches are merged via concatenation. Finally, an MLP with the softmax function is applied to this combined output to generate the final weight predictions. This design ensures a balanced consideration of both structural and feature-based heuristics. Formally, the gating function is defined as follows:

$$G(\mathbf{x}_{ij}, \mathbf{s}_{ij}) = \text{softmax}\Big( f\big(f(\mathbf{x}_{ij}) \| f(\mathbf{s}_{ij})\big)\Big), \tag{2}$$

where $f$ is a MLP and $\|$ denotes the concatenation operator.

## 4.3 Optimization of Link-MoE

Given the experts chosen in Section 5.1, there are several training strategies used by MoE models to combine them. This includes end-to-end training [42] and the EM algorithm [45]. Given the multiple options, we are then tasked with the question: *how do we optimize Link-MoE?*

In this work, we employ a two-step training strategy, as detailed in Algorithm 1 in Appendix E. Initially, we train each expert individually using their respective optimal hyperparameters and perform inference on the dataset to obtain the prediction scores for each link. Subsequently, we focus on the training of the gating model. Specifically, we adopt the cross entropy to train the gating function:

$$L = - \sum_{(i,j) \in \mathcal{P} \bigcup \mathcal{N}} y_{ij} \log Y_{ij} + (1 - y_{ij}) \log(1 - Y_{ij})), \quad (3)$$

where $y_{ij} = 1$ when a link exists between node $v_i$ and $v_j$ in the graph, and $y_{ij} = 0$ otherwise. $\mathcal{P}$ and $\mathcal{N}$ denote the set of positive/negative links in the graph, respectively.

There are several benefits to our two-step training strategy. **(1)** *Efficiency*: This approach eliminates the need to load every expert into memory simultaneously, as each expert is trained and infers independently; For time efficiency, tuning the gating model to identify the best hyperparameters is more efficient since it involves training only MLPs. **(2)** *Effectiveness*: The two-step training strategy helps to avoid the 'collapse problem' often encountered in MoE models [37]. This issue arises when only a single expert is consistently selected, leading to the under-utilization and inadequate learning of the other experts. By training the experts individually first, we mitigate this risk, ensuring a more balanced and effective utilization of all experts. We compare two-step and end-to-end training strategies in Appendix G. **(3)** *Flexibility*: When introducing new experts into the Link-MoE, it's only necessary to train these new experts and the gating model. All previously trained experts can be seamlessly integrated without the need for retraining.

## 5 Experiments

In this section, we conduct comprehensive experiments to validate the effectiveness of the proposed Link-MoE. Specifically, we aim to address the following research questions: **RQ1:** How does Link-MoE perform when compared to other baseline models? **RQ2:** Are the heuristics effective in aiding the selection of experts? **RQ3:** Can Link-MoE adaptively select suitable experts for different node pairs?

### 5.1 Experimental Settings

**Datasets**. We evaluate our proposed method on eight datasets including homophilous graphs: Cora, Citeseer, Pubmed [46], ogbl-ppa, ogbl-collab, and ogbl-citation2 [47] and heterophilic graphs: Chameleon and Squirrel [48]. Please see Appendix A for more details on each dataset.

**Baselines**. We consider a diverse set of baselines include heuristics, embedding methods, GNNs, and GNN4LP methods. This includes: CNs [10], AA [18], RA [19], Shortest Path [43], Katz [11], Node2Vec [49], Matrix Factorization (MF) [50], MLP, GCN [23], GAT [51], SAGE [24], GAE [25], SEAL [15], BUDDY [26], Neo-GNN [14], NCN and NCNC [13], NBFNet [16], PEG [52], LP-Former [27].

Additionally, to comprehensively evaluate the proposed Link-MoE, which integrates multiple link predictors, we design two ensemble baseline methods for comparison. The first method, **Mean-Ensemble**, combines all expert models with uniform weight, ensuring each expert contributes equally to the final prediction. The second method, **Global-Ensemble**, learns a global weight for each expert that is applied when predicting all node pairs. In this approach, each expert contributes to the final prediction based on the learned global weight for all node pairs, allowing for a differentiated influence of each expert based on their performance. See Appendix A.2 for more details on these two methods. Additionally, we compare our method with two other ensemble methods [30, 31], with the results provided in Appendix H.

**Link-MoE Settings**. In this study, we incorporate both node features and a variety of different heuristics as input features for the gating model. These heuristics include node degree, CN, AA, RA, Shortest Path, Katz, and Personalized PageRank (PPR) [20]. Furthermore, our approach uses a wide range of experts, including NCN, NCNC, Neo-GNN, BUDDY, MLP, Node2Vec, SEAL, GCN, NBFNet, and PEG. NBFNet and PEG are only used for smaller datasets, as they often run into out-of-memory issues on the larger OGB datasets. For the two baselines, Mean-Ensemble and Global-Ensemble, we use the same experts as with Link-MoE. More setting details are in Appendix A.2. For evaluation, we report several ranking metrics including the Hits@K and Mean Reciprocal Rank (MRR). In the main paper, we report the MRR for Cora, Citeseer, and Pubmed and for OGB we

use the evaluation metric used in the original study [47]. Results for other metrics are shown in Appendix C.

Table 2: Main results on link prediction (%). Highlighted are the results ranked **first**, **second**, and **third**. We use * to highlight the experts we used in Link-MoE. Notably, NBFNet and PEG are not used as experts on OGB datasets due to their OOM issues.

| | Metric | Cora MRR | Citeseer MRR | Pubmed MRR | ogbl-collab Hits@50 | ogbl-ppa Hits@100 | ogbl-citation2 MRR |
|---|---|---|---|---|---|---|---|
| Heuristic | CN | 20.99 | 28.34 | 14.02 | 61.37 | 27.65 | 74.3 |
| | AA | 31.87 | 29.37 | 16.66 | 64.17 | 32.45 | 75.96 |
| | RA | 30.79 | 27.61 | 15.63 | 63.81 | 49.33 | 76.04 |
| | Shortest Path | 12.45 | 31.82 | 7.15 | 46.49 | 0 | >24h |
| | Katz | 27.4 | 38.16 | 21.44 | 64.33 | 27.65 | 74.3 |
| Embedding | Node2Vec* | 37.29 ± 8.82 | 44.33 ± 8.99 | 34.61 ± 2.48 | 49.06 ± 1.04 | 26.24 ± 0.96 | 45.04 ± 0.10 |
| | MF | 14.29 ± 5.79 | 24.80 ± 4.71 | 19.29 ± 6.29 | 41.81 ± 1.67 | 28.4 ± 4.62 | 50.57 ± 12.14 |
| | MLP* | 31.21 ± 7.90 | 43.53 ± 7.26 | 16.52 ± 4.14 | 35.81 ± 1.08 | 0.45 ± 0.04 | 38.07 ± 0.09 |
| GNN | GCN* | 32.50 ± 6.87 | 50.01 ± 6.04 | 19.94 ± 4.24 | 54.96 ± 3.18 | 29.57 ± 2.90 | 84.85 ± 0.07 |
| | GAT | 31.86 ± 6.08 | 48.69 ± 7.53 | 18.63 ± 7.75 | 55.00 ± 3.28 | OOM | OOM |
| | SAGE | 37.83 ± 7.75 | 47.84 ± 6.39 | 22.74 ± 5.47 | 59.44 ± 1.37 | 41.02 ± 1.94 | 83.06 ± 0.09 |
| | GAE | 29.98 ± 3.21 | 63.33 ± 3.14 | 16.67 ± 0.19 | OOM | OOM | OOM |
| GNN4LP | SEAL* | 26.69 ± 5.89 | 39.36 ± 4.99 | 38.06 ± 5.18 | 63.37 ± 0.69 | 48.80 ± 5.61 | 86.93 ± 0.43 |
| | BUDDY* | 26.40 ± 4.40 | 59.48 ± 8.96 | 23.98 ± 5.11 | 64.59 ± 0.46 | 47.33 ± 1.96 | 87.86 ± 0.18 |
| | Neo-GNN* | 22.65 ± 2.60 | 53.97 ± 5.88 | 31.45 ± 3.17 | 66.13 ± 0.61 | 48.45 ± 1.01 | 83.54 ± 0.32 |
| | NCN* | 32.93 ± 3.80 | 54.97 ± 6.03 | 35.65 ± 4.60 | 63.86 ± 0.51 | 62.63 ± 1.15 | 89.27 ± 0.05 |
| | NCNC* | 29.01 ± 3.83 | 64.03 ± 3.67 | 25.70 ± 4.48 | 65.97 ± 1.03 | 62.61 ± 0.76 | 89.82 ± 0.43 |
| | NBFNet* | 37.69 ± 3.97 | 38.17 ± 3.06 | 44.73 ± 2.12 | OOM | OOM | OOM |
| | PEG* | 22.76 ± 1.84 | 56.12 ± 6.62 | 21.05 ± 2.85 | 49.02 ± 2.99 | OOM | OOM |
| | LPFormer | 39.42 ± 5.78 | 65.42 ± 4.65 | 40.17 ± 1.92 | 68.14 ± 0.51 | 63.32 ± 0.63 | 89.81 ± 0.13 |
| Ensemble | Mean-Ensemble | 39.74 ± 4.70 | 53.73 ± 2.83 | 38.54 ± 5.40 | 66.82 ± 0.40 | 26.70 ± 3.92 | 89.55 ± 0.55 |
| | Global-Ensemble | 38.13 ± 4.60 | 53.96 ± 2.79 | 37.63 ± 6.54 | 67.08 ± 0.34 | 60.67 ± 1.44 | 90.72 ± 0.72 |
| | Link-MoE | 44.03 ± 2.28 | 67.49 ± 0.30 | 53.10 ± 0.24 | 71.32 ± 0.99 | 69.39 ± 0.61 | 91.25 ± 0.02 |
| | Improv. | 10.80% | 3.16% | 18.71% | 4.67% | 9.59% | 0.58% |

## 5.2 Main Results

We present the main results of link prediction for the small datasets and OGB datasets in Table 2. Reported results are mean and standard deviation over 10 seeds. We use "Improv." to denote the relative improvement of Link-MoE over the second best model. From the table, we can have the following observations:

- Link-MoE consistently outperforms all the baselines by a significant margin. For example, it achieves a relative improvement of 18.71% on the MRR metric for the Pubmed dataset and 9.59% on the Hit@100 metric for the ogbl-ppa dataset, compared to the best-performing baseline methods.

- While both the Mean-Ensemble and Global-Ensemble methods also incorporate all the experts used in Link-MoE, their performance is generally subpar in most cases. Although the Global-Ensemble, which learns different weights for each expert, usually outperforms the Mean-Ensemble, it still falls short of the performance of single baseline methods in some scenarios. We attribute this to their inability to adaptively apply different experts to specific node pairs, which demonstrates the effectiveness of the gating model in Link-MoE.

- We further compare against LPFormer [27], a recent method that attempts to adaptively customize pairwise information to each node pair, resulting in strong performance. We find that our model is able to considerably outperform LPFormer on all datasets. From this we conclude that Link-MoE is better than LPFormer at customizing the pairwise information to each node pair.

Additionally, instead of using all experts, we conducted experiments with only a few experts (i.e., 3 or 4 experts). Furthermore, we also explored a sparse gating strategy [37], which selectively activates only the Top-K experts for each sample's prediction. The results, presented in Appendix D, demonstrate that these two variants can achieve comparable performance with using all the experts.

We further evaluate the proposed Link-MoE on heterophilic graphs, with results in Appendix F indicating that Link-MoE achieves strong performance. Additionally, we assess Link-MoE in the more challenging HeaRT setting [53], with results in Appendix J further validating its effectiveness.

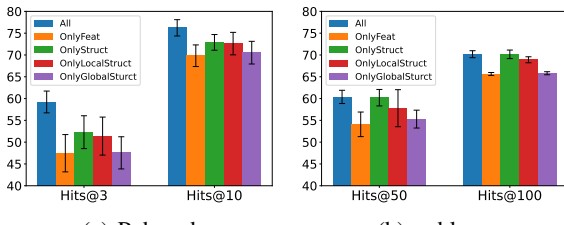

(a) Pubmed       (b) ogbl-ppa

Figure 6: Performance of Link-MoE variants.

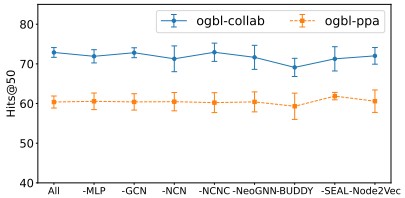

Figure 7: Performance impact of removing a single expert.

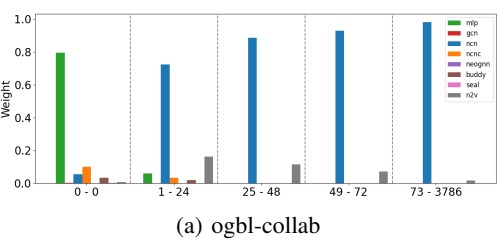

(a) ogbl-collab

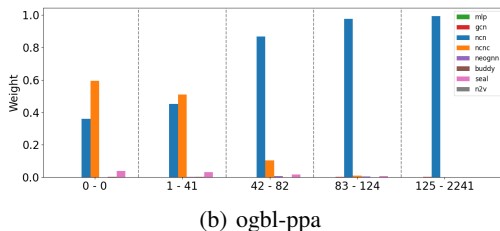

(b) ogbl-ppa

Figure 8: Expert weights for ogbl-collab and ogbl-ppa dataset. The groups are split based on CN.

## 5.3 The Effectiveness of Different Heuristics in Gating

The addition of the gating model enables Link-MoE to significantly surpass both the Mean-Ensemble and Global-Ensemble methods. This improvement underscores the pivotal role of the gating model, which intelligently leverages heuristic information to enhance the overall performance. In this subsection, we delve into the impact of various heuristics utilized by the gating model in Link-MoE. Specifically, there are three types of heuristics: local structure proximity (e.g., CN, AA, RA), global structure proximity (e.g., Shortest Path, Katz), and feature proximity. We design experiments to isolate the effect of specific types of heuristics by including them either individually or in groups. For instance, utilizing only feature proximity is referred to as 'OnlyFeat', combining local and global structure proximities is labeled 'OnlyStruct', employing only local structure proximity is denoted 'OnlyLocalStruct', and using only global structure proximity is marked 'OnlyGlobalStruct'. Additionally, the 'All' is used to indicate leveraging all heuristics. The results on Pubmed and ogbl-ppa datasets are shown in Figure 6. From the experimental results, several observations can be made: **(1)** Using only the feature proximity (OnlyFeat) or only the global structural proximity (OnlyGlobalStruct) typically yields lower performance. **(2)** Combining the local and global structure proximity (OnlyStruct) tends to outperform using either one alone. **(3)** The effectiveness of feature proximity varies between datasets. On Pubmed, it is beneficial to include it but not on ogbl-ppa. This discrepancy could stem from the relative importance of feature information in each dataset, as suggested by the very poor performance of MLP on ogbl-ppa, whose Hits@100 is only around 0.45. Moreover, we explore different inputs for the gating model, the results are presented in Appendix I, which highlights the rationality of the design of our gating model.

## 5.4 The Importance of Different Experts

In the previous sections, we incorporated all the selected experts in Link-MoE. This subsection aims to explore the impact of each expert's removal on the framework's performance. We conducted experiments on the ogbl-collab and ogbl-ppa datasets, utilizing Hits@50 as the metric for evaluation. The results are presented in Figure 7, where the x-axis indicates the experts removed for each trial. We observe that the removal of individual experts does not significantly impact the performance of Link-MoE on both the ogbl-collab and ogbl-ppa datasets. This phenomenon suggest that Link-MoE possesses an adaptive capability to compensate for the absence of certain models by effectively utilizing other available experts.

## 5.5 Analysis of the Gating Weights

In this subsection, we explore the mechanism by which the gating model allocates weights to experts based on the heuristics of different node pairs. Similar to Section 3, we categorize the test node pairs into distinct groups according to different heuristics. Subsequently, for each group, we compute the average weights assigned to each expert by the gating model. The results based on Common Neighbors for ogbl-collab and ogbl-ppa are shown in Figure 8. We can have some interesting findings: **(1)**: For the ogbl-collab dataset, when the node pair doesn't have common neighbor, the gating model usually assigns a large weight to the MLP model, which aligns the analysis in Section 3 that the MLP can perform well when there is no CN on ogbl-collab. However, For the ogbl-ppa dataset, when there is no common neighbor, the gating model would assign a high weight to NCNC. This preference arises because node features hold less significance in the ogbl-ppa dataset and NCNC can leverage multi-hop common neighbor information. **(2)**: As the number of common neighbors increases, the gating model increasingly allocates more weight to the NCN. This shift reflects the NCN's capability to efficiently capture and utilize common neighbor information. **(3)**: Not every expert model contributes much to the prediction. This selective engagement is attributed to the overlapping capabilities of certain models. For instance, both Neo-GNN and NCNC are capable of exploiting multi-hop information. In such cases, the gating model opts for one over the other to avoid redundancy and optimize prediction efficacy. This phenomenon is also consistent with the results in Section 5.4 that removing one expert doesn't affect the overall performance. The analysis of gating weights on heterophilic datasets can be found in Appendix F.

While Link-MoE can successfully leverage heuristics to select appropriate experts for different node pairs, we recognize substantial room for further enhancements. For instance, the observed overlapping ratio between Neo-GNN and NCNC on the ogbl-collab dataset is not very high, as shown in Figure 3, even if they exploit similar heuristics. But in Link-MoE, Neo-GNN has very low weights on the ogbl-collab dataset. This observation underscores the vast potential for advancing MoE applications in link prediction, a direction we intend to explore in future work.

## 6 Conclusion

In this study, we explored various heuristics and GNN4LP models for link prediction. Based on the analysis, a novel MoE model Link-MoE is designed to capitalize on the strengths of diverse expert models for link prediction. The extensive experiments underscore the exceptional performance of Link-MoE in link prediction, validating the rationale behind its design. Furthermore, we also showcase the substantial potential of MoE models for link prediction.

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

# A  Datasets and Experimental Settings

## A.1  Datasets

The statistics for each dataset is shown in Table 3. We adopt the single fixed train/validation/test split with percentages 85/5/10% for Cora, Citeseer, and Pubmed as used in [53]. For the OGB datasets, we use the fixed splits provided by the OGB benchmark [54]. Note that we omit ogbl-ddi due to observations made by Li et al. [53] showing a weak correlation between validation and test performance. For heterophilic graphs Chameleon and Squirrel, we use the same split ratio as Cora.

Table 3: Statistics of datasets. The split ratio is for train/validation/test.

|  | Cora | Citeseer | Pubmed | ogbl-collab | ogbl-ppa | ogbl-citation2 | Chameleon | Squirrel |
|---|---|---|---|---|---|---|---|---|
| #Nodes | 2,708 | 3,327 | 18,717 | 235,868 | 576,289 | 2,927,963 | 2,277 | 5,201 |
| #Edges | 5,278 | 4,676 | 44,327 | 1,285,465 | 30,326,273 | 30,561,187 | 36,101 | 217,037 |
| Mean Degree | 3.9 | 2.81 | 4.74 | 10.90 | 105.25 | 20.88 | 31.71 | 83.46 |
| Split Ratio | 85/5/10 | 85/5/10 | 85/5/10 | 92/4/4 | 70/20/10 | 98/1/1 | 85/5/10 | 85/5/10 |

## A.2  Experimental Settings

**Training Settings**. We use the binary cross entropy loss to train each model. The loss is optimized using the Adam optimizer [55]. At first, we train all of the expert models by using the hyperparameters suggested in this repository *. We then do the inference to obtain the prediction score for each link. Secondly, in order to train Link-MoE, we split the original validation dataset into a new training set and validation set. Thirdly, we train the gating model until it converges and choose the model weights associated with the best validation performance. The rationale for utilizing a portion of the validation set to train the gating model is as follows: Given that each expert model is finely tuned on the training data, there exists a significant disparity between the prediction scores for positive and negative edges within this dataset. Should the original training set be employed to train our gating model, the outcome would be skewed—regardless of the gating model's outputs, predicted scores for positive pairs would invariably remain substantially higher than those for negative pairs. Therefore, to train the gating model effectively, we repurpose the original validation set as our training data, dividing it into new training and validation subsets. Unlike baseline models, which are trained exclusively on the original training set, our method benefits from incorporating a very small validation set compared to the training set into the training process for the gating model, yielding notable performance improvements. Despite this modification, the comparison remains relatively fair. The original validation set is relatively small, so using a portion of it for training does not significantly alter the amount of data available. Thus, the baseline models remain essentially equivalent, even with this adjustment. Moreover, it is common for validation set is to be used for the search of model architectures in neural architecture search. We train both experts and gating models on NVIDIA RTX A6000 GPU with 48GB memory.

**Training Data Split**. In the original datasets, we are given a fixed train, validation and test dataset. These splits are used to train each individual expert. Once each expert model is fully trained, we perform inference on the validation and test sets, thereby obtaining the prediction score for each link. As noted earlier, we split the original validation set into a new training and validation split for training and validating Link-MoE. Note that the original test dataset is still used solely for testing. When splitting the validation set, we use different ratios for different datasets. For ogbl-citation2, ogbl-ppa, ogbl-collab, Citeseer, Chameleon and Squirrel, the ratios are $0.8$, and for Cora and Pubmed, the ratios aer $0.9$.

**Hyperparameter Settings**. The hyperparameter ranges are shown in Table 4. Since our gating model has a low complexity, we can efficiently search over a large hyperparameter space on all eight datasets.

**Evaluation Setting**. The evaluation is conducted by ranking each positive sample against a set of negative samples. The same set of negatives are shared among all positive samples with the exception of ogbl-citation2, which customizes 1000 negatives to each positive sample. The set of negative samples are fixed and are taken from Li et al. [53] and Hu et al. [47] for their respective datasets.

---

*https://github.com/Juanhui28/HeaRT/tree/master

Table 4: Hyperparameter Search Ranges

| Dataset | Learning Rate | Dropout | Weight Decay | # Model Layers | Hidden Dim |
|---|---|---|---|---|---|
| Cora | (0.001, 0.0001, 0.0001) | (0, 0.3, 0.5, 0.8) | (1e-2, 1e-4, 1e-7, 0) | (1, 2, 3) | (8, 16, 32, 64) |
| Citeseer | (0.001, 0.0001, 0.0001) | (0, 0.3, 0.5, 0.8) | (1e-2, 1e-4, 1e-7, 0) | (1, 2, 3) | (8, 16, 32, 64) |
| Pubmed | (0.001, 0.0001, 0.0001) | (0, 0.3, 0.5, 0.8) | (1e-2, 1e-4, 1e-7, 0) | (1, 2, 3) | (8, 16, 32, 64) |
| ogbl-collab | (0.01, 0.001, 0.0001) | (0, 0.3, 0.5) | (1e-7, 0) | (2, 3, 4) | (32, 64, 128) |
| ogbl-ppa | (0.01, 0.001, 0.0001) | (0, 0.3, 0.5) | (1e-7, 0) | (2, 3, 4) | (32, 64, 128) |
| ogbl-citation2 | (0.01, 0.001, 0.0001) | (0, 0.3, 0.5) | (1e-7, 0) | (2, 3, 4) | (32, 64, 128) |
| Chameleon | (0.001, 0.0001) | (0, 0.3, 0.5) | (1e-4, 1e-7, 0) | (1, 2, 3) | (8, 16, 32, 64) |
| Squirrel | (0.01, 0.001) | (0, 0.3, 0.5) | (1e-7, 0) | (2, 3, 4) | (32, 64, 128) |

**Mean-Ensemble**. After training the experts, we can obtain the prediction score on the test set from each expert. Then we take the mean across all experts to get a final score for link in the test set. The purpose of this model is to ascertain whether naively combining the different experts is itself enough for good performance. More formally, the Mean-Ensemble can be defined as follows:

$$Y_{ij} = \frac{1}{m}\sum_{o=1}^{m} E_o(\mathbf{A}, \mathbf{X})_{ij}. \tag{4}$$

**Global-Ensemble**. For the Global-Ensemble, we learn a weight vector $\mathbf{w} = [w_1, w_2, ..., w_m]$ to combine the experts which can be defined as follows:

$$Y_{ij} = \sigma\Big(\sum_{o=1}^{m} w_o E_o(\mathbf{A}, \mathbf{X})_{ij}\Big) \tag{5}$$

Notably, the $\mathbf{w}$ is uniform for all node pairs. Therefore, this method is not able to flexibility adjust the weight of different experts to each node pair. Rather, a single weight is used across all samples. The purpose of this method is to test whether it is necessary to customize the weight of the experts to each link as is done in Link-MoE.

# B  Additional Results for the Preliminary Study

In this section, we extend the preliminary study (Section 3) with additional results, including model and heuristic overlaps on Cora, Citeseer, Pubmed, and ogbl-ppa datasets. We also analyze the performance of various models across node pair groups categorized by Commen Neighbors (CN), Shortest Path (SP), and Feature Cosine Similarity (FCS).

## B.1  Model Overlapping Results

Following Section 3, for the Cora, CiteSeer, PubMed, we use Hits@3. And we use Hits@20 as the metric for ogbl-ppa dataset. The results are shown in Figure 9 and Figure 10. From the results, we can have the following findings:

- For most datasets, the overlapping between different GNN4LP models is not very high, which consistent with the finding in Section 3.
- The feature cosine similarity usually has much smaller overlapping with other methods.

## B.2  The Performance of Different Models on Different Heuristic Groups

In this section, we showcase the varied performances of GNN4LP models across groups categorized by shortest path and feature cosine similarity within the Cora and ogbl-collab datasets. The results for these heuristics are depicted in Figure 11 for shortest path groups and in Figure 12 for cosine similarity groups. We have the following observations:

- For the Cora dataset, SEAL excels over other models at shorter path lengths, while MLP shows superior performance for longer paths. In the ogbl-collab dataset, BUDDY is effective at shorter distances, whereas Neo-GNN, which integrates multi-hop common neighbor information, performs better as the shortest path (SP) length increases.

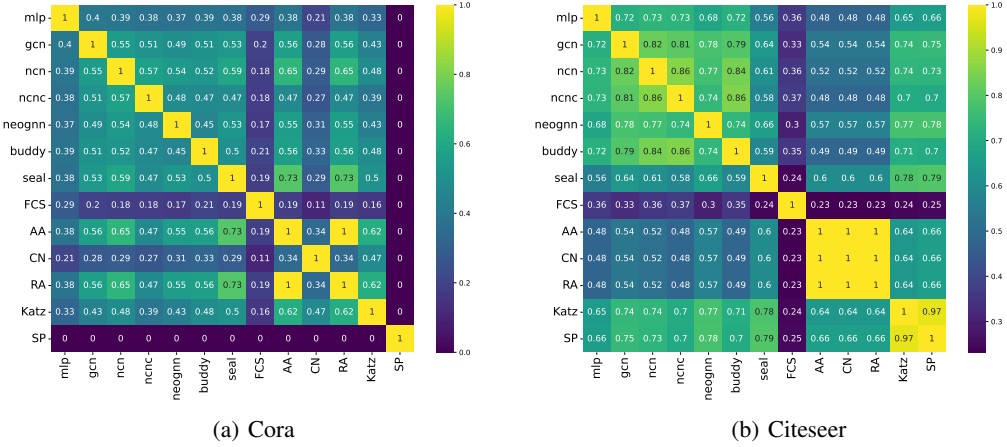

(a) Cora                                      (b) Citeseer

Figure 9: The overlapping ratio of different methods on Cora and CiteSeer dataset.

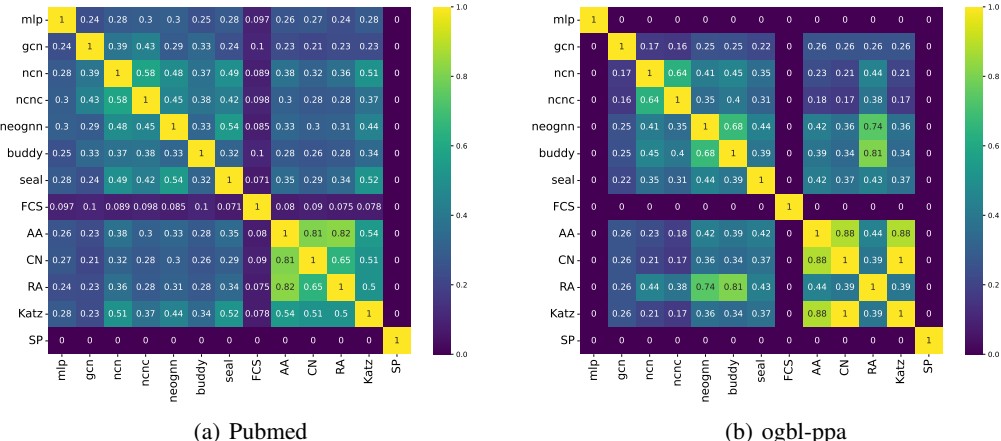

(a) Pubmed                                    (b) ogbl-ppa

Figure 10: The overlapping ratio of different methods on Pubmed and ogbl-ppa dataset.

- For the Cora dataset, the seal excels over other models at smaller feature cosine similarity, while MLP outperforms when the feature cosine similarity is high. For the ogbl-collab dataset, NCNC is effective at low feature cosine similarity, but BUDDY can works well when the feature similarity is high.

- Different datasets exhibit distinct patterns, even within the same heuristic groups. There is no single model consistently outperforms others across all groups.

These observations further validate the rationale behind employing heuristics as inputs to the gating model, effectively leveraging the strengths of various GNN4LP models.

## C   Additional Results on Benchmark Datasets

We present additional results of Cora, Citeseer, Pubmed, and OGB datasets in Table 5, 6, 7, and 8, respectively. We use">24h" to denote methods that require more than 24 hours for either training one epoch or evaluation. OOM indicates that the algorithm requires over 50Gb of GPU memory [53]. Note that LPFormer [27] is omitted from these tables as it doesn't report the results on these additional metrics. The data presented in these tables indicate that Link-MoE consistently achieves the best results in most cases, demonstrating its superior effectiveness in link prediction task. Although it does not always achieve the best performance across all metrics for the Cora and Citeseer datasets, it usually achieves the second or third best results. This slightly diminished performance can likely be

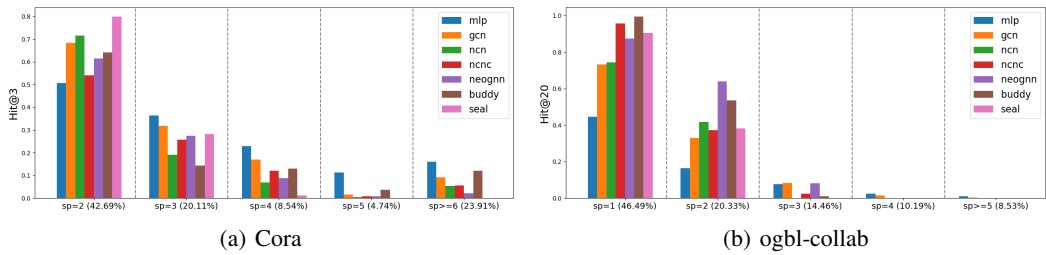

(a) Cora            (b) ogbl-collab

Figure 11: The performance of different models on each shortest path group.

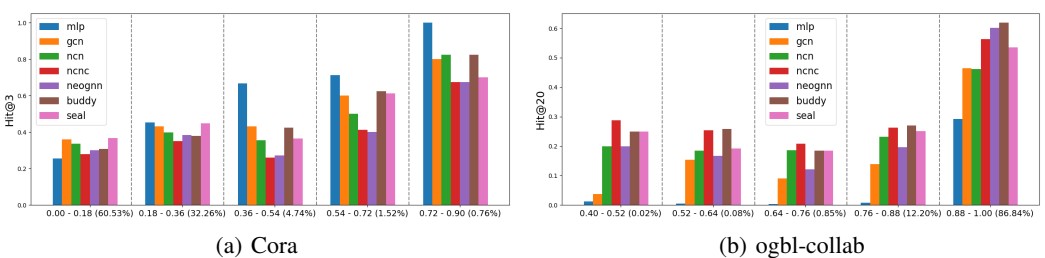

(a) Cora            (b) ogbl-collab

Figure 12: The performance of different models on each feature cosine similarity group.

attributed to the limited number of validation edges available in these datasets, i.e., 263 in Cora and 227 in Citeseer, restricting the amount of information Link-MoE can leverage for optimal learning and performance.

Table 5: Additional results on Cora(%). Highlighted are the results ranked **first**, **second**, and **third**. We use * to highlight the experts we used in Link-MoE.

|  | Models | Hits@1 | Hits@3 | Hits@10 | Hits@100 |
|---|---|---|---|---|---|
| Heuristic | CN | 13.47 | 13.47 | 42.69 | 42.69 |
|  | AA | 22.2 | 39.47 | 42.69 | 42.69 |
|  | RA | 20.11 | 39.47 | 42.69 | 42.69 |
|  | Shortest Path | 0 | 0 | 42.69 | 71.35 |
|  | Katz | 19.17 | 28.46 | 51.61 | 74.57 |
| Embedding | Node2Vec* | 22.3 ± 11.76 | **41.63 ± 10.5** | 62.34 ± 2.35 | 84.88 ± 0.96 |
|  | MF | 7.76 ± 5.61 | 13.26 ± 4.52 | 29.16 ± 6.68 | 66.39 ± 5.03 |
|  | MLP* | 18.79 ± 11.40 | 35.35 ± 10.71 | 53.59 ± 3.57 | 85.52 ± 1.44 |
| GNN | GCN* | 16.13 ± 11.18 | 32.54 ± 10.83 | 66.11 ± 4.03 | 91.29 ± 1.25 |
|  | GAT | 18.02 ± 8.96 | **42.28 ± 6.37** | 63.82 ± 2.72 | 90.70 ± 1.03 |
|  | SAGE | **29.01 ± 6.42** | **44.51 ± 6.57** | 63.66 ± 4.98 | 91.00 ± 1.52 |
|  | GAE | 17.57 ± 4.37 | 24.82 ± 4.91 | 70.29 ± 2.75 | 92.75 ± 0.95 |
| GNN4LP | SEAL* | 12.35 ± 8.57 | 38.63 ± 4.96 | 55.5 ± 3.28 | 84.76 ± 1.6 |
|  | BUDDY* | 12.62 ± 6.69 | 29.64 ± 5.49 | 59.47 ± 5.49 | 91.42 ± 1.26 |
|  | Neo-GNN* | 4.53 ± 1.96 | 33.36 ± 9.9 | 64.1 ± 4.31 | 87.76 ± 1.37 |
|  | NCN* | 19.34 ± 9.02 | 38.39 ± 7.01 | 74.38 ± 3.15 | 95.56 ± 0.79 |
|  | NCNC* | 9.79 ± 4.56 | 34.31 ± 8.87 | 75.07 ± 1.95 | 95.62 ± 0.84 |
|  | NBFNet* | **29.94 ± 5.78** | 38.29 ± 3.03 | 62.79 ± 2.53 | 88.63 ± 0.46 |
|  | PEG* | 5.88 ± 1.65 | 30.53 ± 6.42 | 62.49 ± 4.05 | 91.42 ± 0.8 |
|  | Mean-Ensemble | 26.34 ± 7.96 | 38.80 ± 4.92 | **77.12 ± 2.03** | **96.56 ± 0.64** |
|  | Global-Ensemble | 24.5 ± 7.65 | 36.7 ± 6.66 | **76.81 ± 2.14** | **96.9 ± 0.59** |
|  | Link-MoE | **32.12 ± 4.72** | 38.81 ± 1.09 | **75.84 ± 0.28** | **96.26 ± 0.09** |

# D   Results for Top-K Experts and a few Experts

We conduct experiments on the ogbl-collab and Pubmed datasets, we limit the activation to the Top-3 experts. And we also conduct experiments using 3 or 4 experts. Specifically, for the ogbl-collab, we use MLP, NCNC, BUDDY (3 experts) and Neo-GNN (4 experts); for the Pubmed datasets, we use NCN, SEAL, NCN (3 experts) and MLP (4 experts). The performance metrics used to evaluate this approach are also Hits@50 for ogbl-collab and MRR for Pubmed. The result are shown in Table 9.

Table 6: Additional results on Citeseer(%). Highlighted are the results ranked **first**, **second**, and **third**. We use * to highlight the experts we used in Link-MoE.

|  | Models | Hits@1 | Hits@3 | Hits@10 | Hits@100 |
|---|---|---|---|---|---|
| Heuristic | CN | 13.85 | 35.16 | 35.16 | 35.16 |
|  | AA | 21.98 | 35.16 | 35.16 | 35.16 |
|  | RA | 18.46 | 35.16 | 35.16 | 35.16 |
|  | Shortest Path | 0 | 53.41 | 56.92 | 62.64 |
|  | Katz | 24.18 | 54.95 | 57.36 | 62.64 |
| Embedding | Node2Vec* | 30.24 ± 16.37 | 54.15 ± 6.96 | 68.79 ± 3.05 | 89.89 ± 1.48 |
|  | MF | 19.25 ± 6.71 | 29.03 ± 4.82 | 38.99 ± 3.26 | 59.47 ± 2.69 |
|  | MLP* | 30.22 ± 10.78 | 56.42 ± 7.90 | 69.74 ± 2.19 | 91.25 ± 1.90 |
| GNN | GCN* | 37.47 ± 11.30 | 62.77 ± 6.61 | 74.15 ± 1.70 | 91.74 ± 1.24 |
|  | GAT | 34.00 ± 11.14 | 62.72 ± 4.60 | 74.99 ± 1.78 | 91.69 ± 2.11 |
|  | SAGE | 27.08 ± 10.27 | 65.52 ± 4.29 | 78.06 ± 2.26 | 96.50 ± 0.53 |
|  | GAE | **54.06 ± 5.8** | 65.3 ± 2.54 | 81.72 ± 2.62 | 95.17 ± 0.5 |
| GNN4LP | SEAL* | 31.25 ± 8.11 | 46.04 ± 5.69 | 60.02 ± 2.34 | 85.6 ± 2.71 |
|  | BUDDY* | 49.01 ± 15.07 | 67.01 ± 6.22 | 80.04 ± 2.27 | 95.4 ± 0.63 |
|  | Neo-GNN* | 41.01 ± 12.47 | 59.87 ± 6.33 | 69.25 ± 1.9 | 89.1 ± 0.97 |
|  | NCN* | 35.52 ± 13.96 | 66.83 ± 4.06 | 79.12 ± 1.73 | 96.17 ± 1.06 |
|  | NCNC* | **53.21 ± 7.79** | 69.65 ± 3.19 | 82.64 ± 1.4 | **97.54 ± 0.59** |
|  | NBFNet* | 17.25 ± 5.47 | 51.87 ± 2.09 | 68.97 ± 0.77 | 86.68 ± 0.42 |
|  | PEG* | 39.19 ± 8.31 | 70.15 ± 4.3 | 77.06 ± 3.53 | 94.82 ± 0.81 |
|  | Mean-Ensemble | 32.50 ± 6.21 | **70.53 ± 2.84** | **85.45 ± 2.15** | **97.27 ± 0.40** |
|  | Global-Ensemble | 32.66 ± 6.23 | **71.0 ± 3.03** | **85.09 ± 2.11** | **97.39 ± 0.34** |
|  | Link-MoE | **58.50 ± 0.46** | **76.72 ± 0.24** | **82.77 ± 0.19** | 96.44 ± 0.14 |

Table 7: Additional results on Pubmed(%). Highlighted are the results ranked **first**, **second**, and **third**. We use * to highlight the experts we used in Link-MoE.

|  | Models | Hits@1 | Hits@3 | Hits@10 | Hits@100 |
|---|---|---|---|---|---|
| Heuristic | CN | 7.06 | 12.95 | 27.93 | 27.93 |
|  | AA | 12.95 | 16 | 27.93 | 27.93 |
|  | RA | 11.67 | 15.21 | 27.93 | 27.93 |
|  | Shortest Path | 0 | 0 | 27.93 | 60.36 |
|  | Katz | 12.88 | 25.38 | 42.17 | 61.8 |
| Embedding | Node2Vec* | 29.76 ± 4.05 | 34.08 ± 2.43 | 44.29 ± 2.62 | 63.07 ± 0.34 |
|  | MF | 12.58 ± 6.08 | 22.51 ± 5.6 | 32.05 ± 2.44 | 53.75 ± 2.06 |
|  | MLP* | 7.83 ± 6.40 | 17.23 ± 2.79 | 34.01 ± 4.94 | 84.19 ± 1.33 |
| GNN | GCN* | 5.72 ± 4.28 | 19.82 ± 7.59 | 56.06 ± 4.83 | 87.41 ± 0.65 |
|  | GAT | 6.45 ± 10.37 | 23.02 ± 10.49 | 46.77 ± 4.03 | 80.95 ± 0.72 |
|  | SAGE | 11.26 ± 6.86 | 27.23 ± 7.48 | 48.18 ± 4.60 | 90.02 ± 0.70 |
|  | GAE | 1.99 ± 0.12 | 31.75 ± 1.13 | 45.48 ± 1.07 | 84.3 ± 0.31 |
| GNN4LP | SEAL* | **30.93 ± 8.35** | 40.58 ± 6.79 | 48.45 ± 2.67 | 76.06 ± 4.12 |
|  | BUDDY* | 15.31 ± 6.13 | 29.79 ± 6.76 | 46.62 ± 4.58 | 83.21 ± 0.59 |
|  | Neo-GNN* | 19.95 ± 5.86 | 34.85 ± 4.43 | 56.25 ± 3.42 | 86.12 ± 1.18 |
|  | NCN* | 26.38 ± 6.54 | 36.82 ± 6.56 | **62.15 ± 2.69** | 90.43 ± 0.64 |
|  | NCNC* | 9.14 ± 5.76 | 33.01 ± 6.28 | **61.89 ± 3.54** | **91.93 ± 0.6** |
|  | NBFNet* | **40.47 ± 2.91** | **44.7 ± 2.58** | 54.51 ± 0.84 | 79.18 ± 0.71 |
|  | PEG* | 8.52 ± 3.73 | 24.46 ± 6.94 | 45.11 ± 4.02 | 76.45 ± 3.83 |
|  | Mean-Ensemble | 25.75 ± 10.15 | **44.36 ± 4.29** | 58.50 ± 2.58 | **93.07 ± 0.37** |
|  | Global-Ensemble | 24.36 ± 11.01 | 43.3 ± 6.32 | 58.85 ± 3.14 | **93.14 ± 0.36** |
|  | Link-MoE | **45.13 ± 0.38** | **52.57 ± 2.27** | **61.11 ± 1.03** | 90.38 ± 0.24 |

From the results, we can find that only 3 or 4 experts can achieve comparable performance with using all experts. Notably, these results don't use the computationally intensive SEAL for the ogbl-collab datasets. Furthermore, the inclusion of the less effective MLP expert in the Pubmed dataset still results in performance improvement, highlighting the complementary nature of the experts and the effectiveness of the proposed method. From the result, a sparse gating mechanism can also achieve promising performance on both datasets.

# E  Algorithm

The full algorithm is detailed in Algorithm 1. Notably, line 1-3 trains the experts individually. Line 4 performs the inference on links to obtain the prediction scores using the obtained experts and get the heuristic features for each link. Line 5-7 updates the gating model using Equation 3.

Table 8: Additional results on OGB datasets(%). Highlighted are the results ranked **first**, **second**, and **third**. We use * to highlight the experts we used in Link-MoE.

| | ogbl-collab | | ogbl-ppa | | ogbl-citation2 | | |
| --- | --- | --- | --- | --- | --- | --- | --- |
| | Hits@20 | Hits@100 | Hits@20 | Hits@50 | Hits@20 | Hits@50 | Hits@100 |
| CN | 49.98 | 65.6 | 13.26 | 19.67 | 77.99 | 77.99 | 77.99 |
| AA | 55.79 | 65.6 | 14.96 | 21.83 | 77.99 | 77.99 | 77.99 |
| RA | 55.01 | 65.6 | 25.64 | 38.81 | 77.99 | 77.99 | 77.99 |
| Shortest Path | 46.49 | 66.82 | 0 | 0 | >24h | >24h | >24h |
| Katz | 58.11 | 71.04 | 13.26 | 19.67 | 78 | 78 | 78 |
| Node2Vec* | 40.68 ± 1.75 | 55.58 ± 0.77 | 11.22 ± 1.91 | 19.22 ± 1.69 | 82.8 ± 0.13 | 92.33 ± 0.1 | 96.44 ± 0.03 |
| MF | 39.99 ± 1.25 | 43.22 ± 1.94 | 9.33 ± 2.83 | 21.08 ± 3.92 | 70.8 ± 12.0 | 74.48 ± 10.42 | 75.5 ± 10.13 |
| MLP* | 27.66 ± 1.61 | 42.13 ± 1.09 | 0.16 ± 0.0 | 0.26 ± 0.03 | 74.16 ± 0.1 | 86.59 ± 0.08 | 93.14 ± 0.06 |
| GCN* | 44.92 ± 3.72 | 62.67 ± 2.14 | 11.17 ± 2.93 | 21.04 ± 3.11 | 98.01 ± 0.04 | 99.03 ± 0.02 | 99.48 ± 0.02 |
| GAT | 43.59 ± 4.17 | 62.24 ± 2.29 | OOM | OOM | OOM | OOM | OOM |
| SAGE | 50.77 ± 2.33 | 65.36 ± 1.05 | 19.37 ± 2.65 | 31.3 ± 2.36 | 97.48 ± 0.03 | 98.75 ± 0.03 | 99.3 ± 0.02 |
| GAE | OOM | OOM | OOM | OOM | OOM | OOM | OOM |
| SEAL* | 54.19 ± 1.57 | 69.94 ± 0.72 | 21.81 ± 4.3 | 36.88 ± 4.06 | 94.61 ± 0.11 | 95.0 ± 0.12 | 95.37 ± 0.14 |
| BUDDY* | 57.78 ± 0.59 | 67.87 ± 0.87 | 26.33 ± 2.63 | 38.18 ± 1.32 | 97.79 ± 0.07 | 98.86 ± 0.04 | 99.38 ± 0.03 |
| Neo-GNN* | 57.05 ± 1.56 | 71.76 ± 0.55 | 26.16 ± 1.24 | 37.95 ± 1.45 | 97.05 ± 0.07 | 98.75 ± 0.03 | 99.41 ± 0.02 |
| NCN* | 50.27 ± 2.72 | 67.58 ± 0.09 | 40.29 ± 2.22 | 53.35 ± 1.77 | 97.97 ± 0.03 | 99.02 ± 0.02 | 99.5 ± 0.01 |
| NCNC* | 54.91 ± 2.84 | 70.91 ± 0.25 | 40.1 ± 1.06 | 52.09 ± 1.99 | 97.22 ± 0.78 | 98.2 ± 0.71 | 98.77 ± 0.6 |
| NBFNet | OOM | OOM | OOM | OOM | OOM | OOM | OOM |
| PEG | 33.57 ± 7.40 | 55.14 ± 2.10 | OOM | OOM | OOM | OOM | OOM |
| Mean-Ensemble | 57.96 ± 0.74 | 71.65 ± 0.30 | 5.22 ± 1.12 | 13.94 ± 4.26 | 98.51 ± 0.03 | 99.21 ± 0.03 | 99.57 ± 0.01 |
| Global-Ensemble | 57.91 ± 1.57 | 71.24 ± 0.72 | 27.3 ± 4.94 | 47.27 ± 5.62 | 98.7 ± 0.04 | 99.37 ± 0.02 | 99.69 ± 0.01 |
| Link-MoE | 63.83 ± 0.65 | 75.16 ± 1.64 | 48.36 ± 1.37 | 59.87 ± 0.80 | 98.59 ± 0.02 | 99.28 ± 0.01 | 99.63 ± 0.02 |

Table 9: Results for using Top-K experts or a few experts on ogbl-collab and Pubmed.

| | ogbl-collab | Pubmed |
| --- | --- | --- |
| Best Expert | 66.13 | 44.73 |
| Top-3 Experts | 71.94 | 51.13 |
| 3 Experts | 71.25 | 52.03 |
| 4 Experts | 72.75 | 52.30 |
| All Experts | 71.32 | 53.10 |

# F    Results for Heterophilic Datasets

We conduct experiments on two widely used heterophilic graphs, i.e., Chameleon and Squirrel. We follow the same setting with [56] and use the MRR for the evaluation metrics. The results are shown in Table 10. From the results, we can find the proposed Link-MoE outperforms the baselines by a large margin. These results demonstrate the proposed Link-MoE works well for both the homophilous and heterophilic graphs.

Table 10: Results on heterophilic datasets. The metric is MRR.

| | Node2Vec | MLP | GCN | BUDDY | Neo-GNN | NCN | NCNC | Link-MoE |
| --- | --- | --- | --- | --- | --- | --- | --- | --- |
| Chameleon | 18.14 | 34.65 | 18.44 | 10.96 | 21.63 | 35.31 | 30.87 | **41.20** |
| Squirrel | 10.60 | 13.66 | 25.22 | 3.80 | 8.05 | 26.97 | 22.25 | **31.98** |

We also analyzed the weights generated by the gating mechanism on the two heterophilic graphs. The results are shown in Figure 13. We observed that feature proximity-based experts, such as MLP and GCN, are rarely employed for both datasets. This is consistent with the characteristics of heterophilic graphs, where connect nodes tend to have dissimilar features. These findings demonstrate the effectiveness of our gating design, as it accurately selects the appropriate experts for different types of graphs.

**Algorithm 1** Two-step Training
___

**input** Input graph $\mathcal{G}$, Node feature $\mathbf{X}$, Expert models $\mathbf{E} = \{E_1, E_2, ..., E_m\}$, Positive set $\mathcal{P}$ and Negative set $\mathcal{N}$
**output** Converged Link-MoE
1: **for** i = 1, 2, ..., $m$ **do**
2:     Train each individual expert $E_i$
3: **end for**
4: Get heuristics and prediction scores for positive and negative links
5: **repeat**
6:     Update gating parameters by optimizing Eq. (3)
7: **until** Gating model converge
___

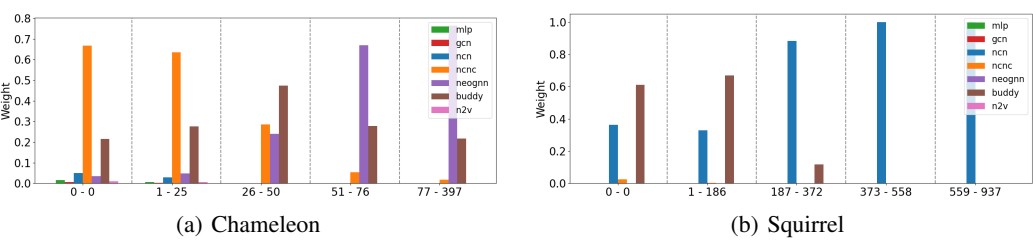

(a) Chameleon          (b) Squirrel

Figure 13: Expert weights for chameleon and squirrel dataset. The groups are split based on CN.

## G    End-to-end Results

In this section, we explore the end-to-end training of the experts and gating models, as shown in Table 11 and Figure 14. Our results show that the convergence speed of different experts varies significantly, which often leads to the model collapsing to a single expert. As a result, the performance of end-to-end training is not as good as the proposed Link-MoE. Despite this, it remains an interesting and challenging idea to explore the effective end-to-end training.

Table 11: Results of the end-to-end training on ogbl-collab and Cora.

|            | ogbl-collab | Cora  |
|------------|-------------|-------|
| end-to-end | 67.43       | 21.97 |
| Link-MoE   | 70.86       | 44.03 |

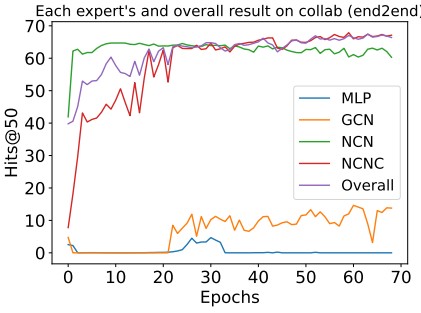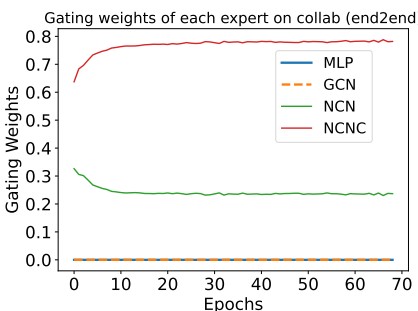

(a) Results of overall & each expert.          (b) The gating weights of each expert.

Figure 14: End-to-end training of experts and gating on ogbl-collab.

## H    Results for Additional Ensemble Methods

We compare our methods with two ensemble methods [30, 31] specifically designed for link prediction tasks. Specifically, [30] found that individual link prediction algorithms exhibit a broad diversity of prediction errors on different graphs and utilized a random forest to ensemble various link predictors; Similarly, [31] employs different graph embedding techniques to train multiple link predictors,

using the outputs of these predictors as inputs for training a subsequent link predictor. We conduct experiments on ogbl-collab and Pubmed dataset. The evaluation metrics are Hits@50 and MRR for ogbl-collab and Pubmed, respectively. The results, shown in Table 12, demonstrate that the proposed Link-MoE model significantly outperforms the ensemble-based methods by a large margin. This can be attributed to the dynamic nature of our gating module, which assigns customized weights to each expert for every node pair.

Table 12: Results for ensemble baselines on ogbl-collab and Pubmed.

|  | ogbl-collab | Pubmed |
| --- | --- | --- |
| Best Expert | 66.13 | 44.73 |
| Mean-Ensemble | 66.82 | 38.54 |
| Global-Ensemble | 67.08 | 37.63 |
| Ensemble [30] | 69.65 | 41.21 |
| Ensemble [31] | 65.11 | 43.92 |
| Link-MoE | 71.32 | 53.10 |

## I   Results for Different Gating Input

We compare our Link-MoE with different gating inputs on ogbl-collab and Pubmed datasets. The evaluation metrics are Hits@50 and MRR for ogbl-collab and Pubmed, respectively. Specifically, we design two different gating inputs.

The first one is traditional gating, which only leverages the node features in gating. The results are shown in Table 13 (Traditional Gating). Traditional Gating only results in comparable performance to the best single experts, while our approach yields superior performance. This phenomenon demonstrates the effectiveness and rationality of the designed gating model.

To investigate the impact of involving expert model predictions as input to the gating model, we conducted experiments on the ogbl-collab and Pubmed datasets by concatenating the prediction results of experts with the heuristic features. The results are shown in Table 13 (With Experts as Input). We observed that involving the experts' prediction results as additional input did not lead to improvement. In fact, this approach may result in lower performance compared to using heuristics alone. This phenomenon suggests that the outputs of the expert models may not effectively reflect their importance to specific node pairs, highlighting the effectiveness and rationality of using heuristics as the gating input.

Table 13: Results for different gating inputs on ogbl-collab and Pubmed.

|  | ogbl-collab | Pubmed |
| --- | --- | --- |
| Best Expert | 66.13 | 44.73 |
| Traditional Gating | 66.59 | 42.15 |
| With Experts as Input | 71.04 | 51.36 |
| Link-MoE | 71.32 | 53.10 |

## J   Results for HeaRT Setting

We conducted additional experiments under the HeaRT setting [53] with OGB datasets, and the results are shown in 14. In the HeaRT setting, hard negative samples are selected for positive samples based on specific heuristics, restricting the negatives to include one of the two nodes from the original positive pair. The results in Table 14 demonstrate that Link-MoE still significantly outperforms the best baseline models, even in this more challenging evaluation scenario.

## K   Limitations

In our current work, we explore various heuristics and GNN4LP models for link prediction and demonstrate the potential of MoE models in this context. The training of the gating model in

Table 14: Results on OGB datasets (%) under HeaRT. Highlighted are the results ranked **first**, **second**, and **third**.

| Models | ogbl-collab | | ogbl-ppa | | ogbl-citation2 | |
|---|---|---|---|---|---|---|
| | MRR | Hits@20 | MRR | Hits@20 | MRR | Hits@20 |
| CN | 4.20 | 16.46 | 25.70 | 68.25 | 17.11 | 41.73 |
| AA | 5.07 | 19.59 | 26.85 | 70.22 | 17.83 | 43.12 |
| RA | 6.29 | **24.29** | 28.34 | 71.50 | 17.79 | 43.34 |
| Shortest Path | 2.66 | 15.98 | 0.54 | 1.31 | >24h | >24h |
| Katz | **6.31** | **24.34** | 25.70 | 68.25 | 14.10 | 35.55 |
| Node2Vec | 4.68 ± 0.08 | 16.84 ± 0.17 | 18.33 ± 0.10 | 53.42 ± 0.11 | 14.67 ± 0.18 | 42.68 ± 0.20 |
| MF | 4.89 ± 0.25 | 18.86 ± 0.40 | 22.47 ± 1.53 | 70.71 ± 4.82 | 8.72 ± 2.60 | 29.64 ± 7.30 |
| MLP | 5.37 ± 0.14 | 16.15 ± 0.27 | 0.98 ± 0.00 | 1.47 ± 0.00 | 16.32 ± 0.07 | 43.15 ± 0.10 |
| GCN | 6.09 ± 0.38 | 22.48 ± 0.81 | 26.94 ± 0.48 | 68.38 ± 0.73 | 19.98 ± 0.35 | 51.72 ± 0.46 |
| GAT | 4.18 ± 0.33 | 18.30 ± 1.42 | OOM | OOM | OOM | OOM |
| SAGE | 5.53 ± 0.5 | 21.26 ± 1.32 | 27.27 ± 0.30 | 69.49 ± 0.43 | **22.05 ± 0.12** | **53.13 ± 0.15** |
| GAE | OOM | OOM | OOM | OOM | OOM | OOM |
| SEAL | **6.43 + 0.32** | 21.57 + 0.38 | 29.71 ± 0.71 | 76.77 ± 0.94 | 20.60 ± 1.28 | 48.62 ± 1.93 |
| BUDDY | 5.67+0.36 | 23.35 + 0.73 | 27.70 ± 0.33 | 71.50 ± 0.68 | 19.17 ± 0.20 | 47.81 ± 0.37 |
| Neo-GNN | 5.23 +0.9 | 21.03 + 3.39 | 21.68 ± 1.14 | 64.81 ± 2.26 | 16.12 ± 0.25 | 43.17 ± 0.53 |
| NCN | 5.09 + 0.38 | 20.84 + 1.31 | **35.06 ± 0.26** | **81.89 ± 0.31** | **23.35 ± 0.28** | **53.76 ± 0.20** |
| NCNC | 4.73 + 0.86 | 20.49+3.97 | **33.52 ± 0.26** | **82.24 ± 0.40** | 19.61 ± 0.54 | 51.69 ± 1.48 |
| NBFNet | OOM | OOM | OOM | OOM | OOM | OOM |
| PEG | 4.83 ± 0.21 | 18.29 ± 1.06 | OOM | OOM | OOM | OOM |
| Link-MoE | **15.11 ± 8.28** | **29.80 ± 4.43** | **62.11 ± 3.54** | **88.49 ± 0.56** | **24.07 ± 1.77** | **57.71 ± 0.19** |

Link-MoE relies on pre-trained experts and heuristic features generated from the graphs. Although we can select a few experts, training these pre-trained experts remains time-consuming, especially for complex models. Additionally, since real-world graphs come from diverse domains and the graph generation process might be quite different, the heuristics used in this paper might not be comprehensive to other domains. Future work should focus on exploring more heuristics to better accommodate diverse domains.

## L  Impact Statement

In this paper, we explore the use of a mixture of experts (MoE) model for use in link prediction. We view the impact of this work as positive, as it can help improve performance of link prediction in many real-world applications including drug discovery and recommender systems. In general, we don't envision any specific negative societal consequences of our work. However, it is possible that the choice of experts used in our model may introduce some negative effects that stem from those individual models.

## M  Dataset Licenses

The license for each dataset can be found in Table 15.

Table 15: Dataset Licenses.

| Datasets | Cora | Citeseer | Pubmed | ogbl-collab | ogbl-ppa | ogbl-citation2 | Chameleon | Squirrel |
|---|---|---|---|---|---|---|---|---|
| License | NLM License | NLM License | NLM License | MIT License | MIT License | MIT License | GPLv3 | GPLv3 |

