# OpenReview forum: "Mixture of Link Predictors on Graphs"
_NeurIPS.cc/2024/Conference — NeurIPS 2024 poster_

### Official Review · Reviewer_SDXC · 2024-06-29

**Soundness:** 2
**Presentation:** 4
**Contribution:** 2
**Rating:** 4
**Confidence:** 5

**Summary:**

This paper proposes an ensemble model of different link prediction methods. The authors find that different node pairs on the graph can form a link due to different pairwise representations, and there is no single link prediction model that can capture all of them. Then the author proposes to combine the outputs of different link prediction models into one ensemble model, weighted by the heuristics information of the node pair. Experiments on graph benchmarks show the improvement brought by ensembling.

**Strengths:**

1. The paper is clear and well-written.

2. The motivation of the study is clearly stated. In the preliminary study section, the authors give solid examples of why individual link prediction models can fail, which motivates a combination of such models.

3. The experiment is convincing and comprehensive, showing the competitive performance improvement with a mixture of link prediction models.

**Weaknesses:**

1. The novelty of the paper is limited. This study proposes a mixture of existing link prediction models, which brings minor research values to the community. The overall contribution of the paper makes it more like a technical report, rather than a research paper. While this paper has its own value, it is more suitable to be submitted to other venues.

2. The baselines considered in the experiment are not comprehensive. For example, [1], which is an ensemble of link prediction models, is not evaluated as a baseline.

3. The overall computational efficiency of the mixture method is concerning. On each graph, a set of link prediction models needs to be trained first, which typically involves an extensive process of hyperparameter tuning like in NCN [2]. This step can be too costly for most real-world use cases. The idea of MoE, especially conditional computation, is not leveraged at all to reduce the computational cost.

4. The authors can further benchmark the Link-MoE on [3], which introduces more challenging link prediction tasks.

[1] Ghasemian, A., Hosseinmardi, H., Galstyan, A., Airoldi, E. M., & Clauset, A. (2020). Stacking models for nearly optimal link prediction in complex networks. Proceedings of the National Academy of Sciences.

[2] Wang, X., Yang, H., & Zhang, M. (2023). Neural common neighbor with completion for link prediction.

[3] Li, J., Shomer, H., Mao, H., Zeng, S., Ma, Y., Shah, N., ... & Yin, D. (2024). Evaluating graph neural networks for link prediction: Current pitfalls and new benchmarking. Advances in Neural Information Processing Systems.

**Questions:**

1. For the gating model, only the heuristics information is considered as the input. However, this can limit the expressiveness of gating models, because the inherent limitation of the heuristics models themselves. Do authors consider involving the expert models as inputs to the gating models?

2. Following up on the question above, is there any ablation study on training the gating model and expert models jointly? It may introduce higher memory consumption, but one can consider remove the computational heavy models like SEAL or NBFNet.

**Limitations:**

See the weakness and question parts.

---

> ### Author Rebuttal · Authors · 2024-08-07
>
> Dear Reviewer SDXC,
>
> We appreciate your constructive feedback. We are pleased to provide detailed responses to address your concerns.
>
> **W1:The novelty of the paper is limited.**
>
> **A1:** The novelty of Link-MoE: Our Link-MoE model stands out due to its innovative approach, leveraging the complementary strengths of different GNN4LP models, resulting in substantial performance gains. The core of our model's success lies in the following findings and innovations:
>
> *1. GNN4LP Models are Complementary:* Our preliminary studies show that the overlapping between different GNN4LP models is notably low, shown in Figure 3, 9 and 10 in our paper, indicating these models offer complementary insights for link prediction. Different node pairs may be best addressed by different models. The critical challenge, then, is determining the most effective way to assign node pairs to the appropriate GNN4LP models.
>
> *2. Heuristic-Informed Gating Mechanism:* Our preliminary studies revealed that different GNN4LP models perform differently across different heuristic groups. By designing a gating model that intelligently applies these heuristics, we facilitate the optimal assignment of node pairs to their most suitable predictors, which is a significant departure from traditional MoE applications, which use the same input features for the gating and experts. We compare the Link-MoE with traditional gating, which only leverages the node features in gating. The results are shown in Table 1 (Traditional MoE) in the global response. Traditional MoE only results in comparable performance to the best single experts, while our approach yields superior performance. This phenomenon demonstrates the effectiveness and rationality of the designed gating model.
>
> **W2:The baselines considered in the experiment are not comprehensive.**
>
> **A2:** Thank you for pointing out the related papers. Despite the Mean-Ensemble and Global-Ensemble method in our paper, we test two more ensemble methods. [1] and [2]  utilized a random forest and MLP to ensemble various link predictors, respectively. The results are presented in Table 2 in the global response. Link-MoE outperforms all ensemble methods. The superior performance of our approach is attributed to the dynamic nature of our gating module, which assigns customized weights to each expert for every node pair.
>
> [1] Stacking Models for Nearly Optimal Link Prediction in Complex Networks, PNAS'20
>
> [2] An Ensemble Model for Link Prediction based on Graph Embedding, Decision Support Systems'22
>
> **W3:The overall computational efficiency of the mixture method is concerning. The idea of MoE, especially conditional computation, is not leveraged at all to reduce the computational cost.**
>
> **A3:** We would like to highlight that our framework differs from traditional MoE methods. Our goal is to leverage different GNN4LP models to address various node pairs effectively. We utilize a two-step training strategy to train the experts and a lightweight gating model separately. While training multiple experts does increase computational cost, our approach only needs a few experts to achieve superior performance compared to the baselines.
>
> To demonstrate this, we conducted experiments on ogbl-collab and Pubmed datasets using only 3 or 4 experts. The detailed settings and results are shown in Table 1 (3 Experts \& 4 Experts) in the global response.
>
> We observe that using only 3 or 4 experts can achieve comparable performance to using all experts, indicating the computational cost of Link-MoE is not prohibitively high.
>
> **W4:The authors can further benchmark the Link-MoE on [3].**
>
> **A4:** We conduct experiments on ogbl-collab and PubMed under HeaRT setting [3]. The results shown in Table 4 of global response demonstrate that our model significantly outperforms all individual experts on both datasets. This highlights the effectiveness of our approach in leveraging the strengths of multiple experts.
>
> **Q1:Do authors consider involving the expert models as inputs to the gating models?**
>
> **A5:** We use the heuristic as input because the GNN4LP models are aligned with the heuristic methods, as investigated in Section 3.2 in our paper. To investigate the impact of involving expert model predictions as input to the gating model, we conducted experiments on the ogbl-collab and Pubmed datasets by concatenating the prediction results of experts with the heuristic features. The results are shown in Table 1 (With Experts as Input) of global response.
>
> We observed that involving the experts' prediction results as additional input did not lead to improvement. In fact, this approach may result in lower performance compared to using heuristics alone. This phenomenon suggests that the outputs of the expert models may not effectively reflect their importance to specific node pairs, highlighting the effectiveness and rationality of using heuristics as the gating input.
>
> **Q2:Is there any ablation study on training the gating and expert models jointly?**
>
> **A6:** We explored an end2end training strategy that trains several experts alongside the gating module from scratch. We conducted experiments on Cora and ogbl-collab to evaluate this approach. The results and experimental settings are detailed in Table 3 of global response.
>
> The end2end results were significantly worse than  Link-MoE. A potential reason is that different experts have varying convergence rates, leading to the collapse problem. Figure 1(a) of global response illustrates some strong experts, such as NCN and NCNC, converge faster and dominate the performance. Strong models are assigned higher gating weights while MLP and GCN are assigned zero weights, as shown in Figure 1(b) of global response,  which limits its ability to leverage additional information from MLP and GCN. While our end2end approach did not yield improved results, it remains an interesting and challenging idea to explore the conditional computation of MoE for efficient end2end training.

---

> > ### Comment · Reviewer_SDXC · 2024-08-11
> >
> > Thanks for the extensive efforts in addressing my concerns. However, I still have several major concerns that prevent me from recommending acceptance of the work. As previously discussed in W1 and W3, I am still concerned about the contribution and motivation of Link-MoE.
> >
> > 1. Link-MoE achieves good performance by ensembling a set of SOTA GNN4LP models. However, this makes the contribution limited because ensembling may be the most well-known method to improve performance. In general, it is not a "surprising" thing to see performance gain by bundling SOTA methods together.
> >
> > 2. The MoE naming is a bit misleading. The author introduces the "MoE" techniques in Link-MoE. However, Link-MoE does not either (1) reduce the parameter size of the entire model; or (2) achieve better efficiency by routing the prediction to different domain experts.
> >
> > 3. Given that LP is a practical problem, scalability is one of the most crucial aspects to consider when choosing LP methods. However, Link-MoE is computationally expensive in both the training and inference stages. During training, the expert model needs to be trained individually with proper hyperparameter tuning. (This is acknowledged in the Limitation section of the paper). During the inference, Link-MoE will require the link representation from all the expert models. This paradigm makes Link-MoE almost impossible to be deployed in industrial applications.
> >
> > If possible, I want to know more about the authors’ comments on the practical applicability of Link-MoE to real-world scenarios.
> >
> > 4. As far as I can see from the paper, the only aspect (research value vs novelty vs insights vs practical usage) that Link-MoE shines on is its superior performance improvement on various benchmarks. However, as [3] points out, HeaRT is a more realistic and personalized evaluation benchmark for LP methods. Since evaluating on HeaRT only requires a small change in the model inference stage (change the random negative edges to hard negative edges in HeaRT) and no need to retrain anything, it is expected that Link-MoE can be evaluated on all datasets in HeaRT, especially those OGB datasets. However, the results are missing from the rebuttal or main experiment. When I revisited HeaRT paper, Observation 1 in Section 4.3 claims that simpler models actually perform better than complex ones. This seems to contradict what is observed in Link-MoE.
> >
> > In general, I find the paper well-written and the preliminary study part is interesting. However, the major concerns above outweigh the paper's strength. Therefore, I keep my original rating.

---

> > > ### Author Response · Authors · 2024-08-12
> > > **Further response to Reviewer SDXC - Part 1**
> > >
> > > Dear Reviewer SDXC,
> > >
> > > Thank you for your prompt reply, allowing us the opportunity to further clarify your concerns.
> > >
> > > ## 1. Difference from Traditional Ensembling Methods
> > >
> > > Although Link-MoE leverages different link prediction models, it is not a simple ensembling method. Meanwhile, we respectively disagree "it is not a "surprising" thing to see performance gain by bundling SOTA methods together.".  In fact, traditional ensembling methods often do not outperform the strongest individual base model. To illustrate this, we tested four ensembling methods, including Mean-Ensemble and Global-Ensemble in our paper, as well as two new ensembling methods [1][2]. For all these methods, we used the same base models to ensure a fair comparison. The results are shown below:
> > >
> > > |  | ogbl-collab | Pubmed |
> > > |---|---|---|
> > > | Best Expert | 66.13 | 44.73 |
> > > | Mean-Ensemble | 66.82 | 38.54 |
> > > | Global-Ensemble | 67.08 | 37.63 |
> > > | Ensemble [1] | 69.65 | 41.21 |
> > > | Ensemble [2] | 65.11 | 43.92 |
> > > | Traditional gating | 66.59 | 42.15 |
> > > | Link-MoE | 71.32 | 53.10 |
> > >
> > > The results indicate that these traditional ensembling methods typically have comparable or even worse performance compared to the best individual expert. This underscore the key contribution to Link-MoE’s success is its ability to assign different node pairs to different experts based on our gating mechanism. The design of this gating mechanism is informed by our preliminary studies, which revealed that different experts excel at different heuristics.
> > >
> > > Additionally, we tested the traditional gating in MoE, which uses the same input as the experts. The results show that the performance of 'Traditional Gating' is even worse than the best expert. This highlights that the strength and contribution of Link-MoE lies not in simple ensembling, but in its intelligent and heuristic-informed allocation of node pairs to the most suitable experts.
> > >
> > > ## 2. The name of MoE
> > >
> > > The term "Mixture of Experts" (MoE) is a broad concept that can be traced back to foundational works such as [3][4]. These MoE models follow the divide-and-conquer principle, where the problem space is divided into subspaces, each of which is potentially easier to solve with specialized experts. Many papers [3,4,5,6] have leveraged the MoE framework to improve model performance, as outlined in the survey [7]. In recent years, MoE has gained popularity again due to the introduction of sparse gating mechanisms, which allows for increasing the model parameters while maintaining efficiency. However, the essence of MoE is not limited to parameter efficiency or reduction. The core idea is the strategic use of specialized experts to handle different aspects of the problem. Our Link-MoE model adheres to this divide-and-conquer principle by routing different node pairs to different experts based on the characteristics of the node pairs. Therefore, we believe that Link-MoE can be considered as a MoE model.

---

> > > ### Author Response · Authors · 2024-08-12
> > > **Further response to Reviewer SDXC - Part 2**
> > >
> > > ## 3. The Practical Applicability of Link-MoE.
> > >
> > > Some of our authors have ever worked in industry and we have collaborated with industry on various link prediction problems. Based on our experiences, we are glad to discuss the potential practical applicability of Link-MOE from the following two perspectives:
> > >
> > > First, in many practical link prediction applications, both the training and inference are offline.  For example, in friend recommendations in social media, the potential list of friends for a particular user is often pre-computed. Therefore in these applications, Link-MOE is potentially to be applicable.
> > >
> > > Second, there are also many link prediction applications where we indeed need to do online inference such as these session-based recommendations. For these applications, we would like to highlight that Link-MOE is also potentially to be applicable with the following two reasons.  (1) These applications often adopt a two-stage strategy to ensure the efficiency. They will first use a simple method to recall a small subset of items, say $L$. Then they will leverage link prediction algorithm to only score these $L$ items. $L$ is much smaller than the whole set of items which is often hundreds or thousands. That is why sophisticated methods can be applied in these applications.  (2) We acknowledge that during training, Link-MoE requires the training of several experts, which can be computationally intensive. However, the training phase is often conducted offline, meaning that the computational cost during training is less of an immediate concern.
> > >
> > > **Regarding the inference, our Link-MoE can also leverage the Sparse Gating to improve the efficiency.** During inference, the gating model first determines the importance of each expert, allowing us to calculate predictions using only the most relevant experts. This selective approach significantly reduces the computational load during inference. Here, we demonstrate the results of Top-2 and Top-3 gating, which only use 2 or 3 experts for each sample during inference.
> > >
> > > |  | ogbl-collab | Pubmed |
> > > |---|---|---|
> > > | Best Expert | 66.13 | 44.73 |
> > > | Top-2 Gating | 71.22 | 50.03 |
> > > | Top-3 Gating | 71.94 | 51.13 |
> > > | All Experts | 71.32 | 53.15 |
> > >
> > > From the results, using only 2 or 3 experts can achieve performance comparable to using all experts. This demonstrates the potential of using Link-MoE for online inference in real-world applications.
> > >
> > > ## 4. The HeaRT setting.
> > >
> > > We would like to clarify that the statement "Observation 1 in Section 4.3 claims that simpler models actually perform better than complex ones" is not accurate. Observation 1 in the HeaRT paper actually claims that the performance gap between simpler models and GNN4LP models is reduced under the HeaRT setting. However, GNN4LP models still outperform simpler models, as shown in Table 5 of the latest version of the HeaRT paper[8] (v3 version on arXiv).
> > >
> > > We have conducted additional experiments under the HeaRT setting with all OGB datasets, and the hit@20 results are shown below. Our findings indicate that Link-MoE still outperforms the best baseline models by a significant margin, even in this more challenging evaluation setting.
> > >
> > > |  | ogbl-collab | ogbl-ddi | ogbl-ppa | ogbl-citation2 |
> > > |---|---|---|---|---|
> > > | Best Expert | 23.35 ± 0.73 | 67.19 ± 1.18 | 82.24 ± 0.40 | 53.76 ± 0.20 |
> > > | Link-MoE | 39.58 ± 0.10 | 68.73 ± 0.43 | 88.49 ± 0.56 | 58.04 ± 0.47 |
> > >
> > > [1] Ghasemian, Amir, et al. Stacking Models for Nearly Optimal Link Prediction in Complex Networks, PNAS'20
> > >
> > > [2] Chen, Yen-Liang, Chen-Hsin Hsiao, and Chia-Chi Wu. "An ensemble model for link prediction based on graph embedding." Decision Support Systems 157 (2022): 113753.
> > >
> > > [3] Jacobs, Robert A., et al. "Adaptive mixtures of local experts." Neural computation 3.1 (1991): 79-87.
> > >
> > > [4] Jordan, Michael I., and Robert A. Jacobs. "Hierarchical mixtures of experts and the EM algorithm." Neural computation 6.2 (1994): 181-214.
> > >
> > > [5] Shahbaba, Babak, and Radford Neal. "Nonlinear models using Dirichlet process mixtures." Journal of Machine Learning Research 10.8 (2009).
> > >
> > > [6] Eigen, David, Marc'Aurelio Ranzato, and Ilya Sutskever. "Learning factored representations in a deep mixture of experts." arXiv preprint arXiv:1312.4314 (2013).
> > >
> > > [7] Masoudnia, Saeed, and Reza Ebrahimpour. "Mixture of experts: a literature survey." Artificial Intelligence Review 42 (2014): 275-293.
> > >
> > > [8] Li, Juanhui, et al. "Evaluating graph neural networks for link prediction: Current pitfalls and new benchmarking." Advances in Neural Information Processing Systems 36 (2024).
> > >
> > > We hope that we have addressed the concerns in your comments, and please kindly let us know if there is any further concern, and we are happy to clarify.
> > >
> > > Best regards,
> > >
> > > All authors

---

> > > ### Author Response · Authors · 2024-08-13
> > > **A friendly reminder**
> > >
> > > Dear Reviewer SDXC,
> > >
> > > We appreciate your thorough review and have provided a detailed response to your concerns. Since the end of the discussion period is approaching, we kindly request your feedback on our response. Your insights are crucial, and we want to ensure that any remaining issues are addressed before the discussion period concludes. Please let us know if there are any further points that need clarification.
> > >
> > > Thank you for your time and consideration.
> > >
> > > Best regards,
> > >
> > > All authors

---

> > > > ### Comment · Reviewer_SDXC · 2024-08-14
> > > >
> > > > Thank the authors for addressing my questions. I have further concerns regarding the scalability of Link-MoE:
> > > >
> > > > > The training and inference of LP models can both happen offline.
> > > >
> > > > However, it is not a solid reason that a practical method can trade efficiency for performance gain. One of the most challenging issues of LP tasks is that the potential edges to score is $O(N^2)$. Therefore, even in an offline setting, efficiency is of great concern for any practical LP model.
> > > >
> > > > > Two-stage strategy to recall first and score later.
> > > >
> > > > This is a practical strategy. However, it is unknown whether Link-MoE can still perform well with such a strategy. Most likely, the performance will drop somehow depending on the selected method at the recall stage. On the contrary, experts like NCN or BUDDY can serve as the entire pipeline to score the edges and maintain the performance reported in the paper. Therefore, it is difficult to claim that Link-MoE is still significantly better than the individual experts in the industrial setting. BTW, I have no much worry about the training cost of Link-MoE. But the inference side is of great concern.
> > > >
> > > > > Sparse Gating.
> > > >
> > > > In my opinion, this is what should be implemented for Link-MoE. It can dynamically adjust the computation cost and only run complex LP models when needed. I **strongly encourage** the authors to expand the discussion on this part, including the method description and experiment results. From the experiment so far, it is hard to tell how good this sparsing gate could be compared to other individual methods. If the authors have the chance to revise the paper substantially, this spare gating method should be the major method introduced in the paper, rather than the current one.
> > > >
> > > > > About inference time.
> > > >
> > > > The current Link-MoE includes SEAL as the expert. I wonder why the authors made such a decision, given that SEAL is known to be both expressive and expensive. In [1], SEAL is reported to run for 28 hours on Citation2 for inference only. Then on PPA, the training and testing time will be over 24 hours. However, the paper does not report the Shortest Path on Citation2 because >24h. Didn't that already surpass 24 hours?
> > > >
> > > > > Another minor question
> > > >
> > > > is that I just found both in the code and paper, the authors are aware of the HeaRT setting. Furthermore, the code itself is heavily built on HeaRT. I just wonder why the HeaRT results are not initially included in the main paper since such a more challenging setting can better prove the strength of Link-MoE.
> > > >
> > > > [1] Labeling Trick: A Theory of Using Graph Neural Networks for Multi-Node Representation Learning.

---

> > > > > ### Author Response · Authors · 2024-08-14
> > > > > **Further response to Reviewer SDXC**
> > > > >
> > > > > Dear Reviewer SDXC,
> > > > >
> > > > > Thank you for sharing your additional concerns. We appreciate your thorough review and are happy to provide detailed responses to each of your concerns.
> > > > >
> > > > > ### **1. Offline and Two-stage strategy**
> > > > >
> > > > > We would like to clarify your first and second concerns together as below:
> > > > >
> > > > > We totally agree with you that the time complexity of LP is $O(N^2)$, making it impractical to apply even simple models to rank all possible edges. Based on our industry experience and collaborations on various link prediction problems, (1) most real-world applications of link prediction no matter whether the inference is offline or online adopt a two-stage strategy; and (2) Models like NCN and BUDDY also cannot be directly applied but utilize this two-stage approach. Therefore, Link-MoE has the potential to be applied just like the individual expert (e.g., NCN or BUDDY)
> > > > >
> > > > > **Performance of Link-MoE with the two-stage strategy.** In fact, the HeaRT setting is similar or mimicking the two-stage strategy. As stated in the Section 4.2 of HeaRT paper, *"The key challenge is how to generate a subset of $S(u,a)$. ... This is similar to the concept of candidate generation, which only ranks a subset of candidates that are most likely to be true."* Particularly, in the HeaRT setting, negative samples for each node are first generated based on multiple heuristics, which serves as the first stage to recall a small set of items. Then the evaluated LP method will score this subset which is against the groudtruth to get the metric performance. Therefore, our experimental results under the HeaRT setting, which show that Link-MoE outperforms the best expert by a significant margin on all OGB datasets, can indicate that Link-MoE remains effective even under the two-stage strategy.
> > > > >
> > > > > ### **2. Sparse Gating**
> > > > >
> > > > > Thank you for emphasizing the importance of sparse gating. We agree that sparse gating is crucial for improving the inference efficiency of Link-MoE.  Actually, we have discussed the sparse gating in the Section 5.2 and Appendix D of our paper. Our experimental results on both the ogbl-collab and Pubmed datasets show that even using top-2 gating can outperform the best expert by a large margin. We are currently running additional experiments with sparse gating and will include more discussions and results in the revision, as you suggested.
> > > > >
> > > > > ### **3. About Inference time**
> > > > >
> > > > > Regarding the selection of experts, we did not specifically choose certain experts; rather, we included all popular baselines as experts. Besides, we also conducted experiments with fewer experts, as shown in Appendix D of our paper.  For example, on the ogbl-collab dataset, we used MLP, NCNC, BUDDY (3 experts), and Neo-GNN (4 experts).
> > > > >
> > > > > |  | Best Expert | 3 Experts | 4 Experts | All Experts |
> > > > > |---|---|---|---|---|
> > > > > | Hit@50 | 66.13 | 71.25 | 72.75 | 71.32 |
> > > > >
> > > > > We can find that good performance can still be achieved with 3 or 4 experts without SEAL. Additionally, the Figure 7 in our paper shows that removing any one expert, including SEAL, does not significantly impact performance.
> > > > >
> > > > > For the Citation2 dataset in both the regular and HeaRT settings, we did not use the Shortest Path heuristic. Besides, for all the ogbl dataset in the HeaRT setting, we didn't use the SEAL as the inference time is slow. We will include these details in the revision.
> > > > >
> > > > > ### **4. HeaRT results**
> > > > >
> > > > > We have submitted a paper on LP with the HeaRT setting but were asked to using the traditional setting. It could be because the HeaRT paper is published last year, and most of the papers still adopt the traditional setting. Based on this experience, we initially adopt the traditional setting.  The reason why we heavily rely on the HeaRT implantation is the code provided in the HeaRT paper includes implementations for both the traditional and HeaRT settings.
> > > > >
> > > > > **We really appreciate your comments and follow-ups. This process indeed helps us demonstrate more advantages of Link-MoE, clarify its potential applications, and think about potential future works of Link-MoE. We will include all these discussions in the revision and we look forward to any further comments or questions you may have.**
> > > > >
> > > > > Best regards,
> > > > >
> > > > > All authors

---

> > > > > > ### Comment · Reviewer_SDXC · 2024-08-14
> > > > > >
> > > > > > Thanks for the detailed responses. In light of the authors' effort to address my concerns, I raised my score to 4 for now, I will explain why I made such a decision:
> > > > > >
> > > > > > 1. I still have concerns about the scalability of Link-MoE. I agree with the authors' responses above. However, just as the authors suggested, Link-MoE is **potentially** as scalable as other sota individuals. All the discussions about two-stage/offline are hypothetical. No performance/inference time is measured under this setting. What we can see so far is that Link-MoE is good at performance but very bad at inference time.
> > > > > >
> > > > > > 2. I think Link-MoE will be a much solid study if the main method is the Sparse Gating one. I believe such a design could be much more efficient. An experiment benchmarking time complexity can present how efficient Link-MoE can be. However, the current version of the manuscript and the rebuttal do not include these.
> > > > > >
> > > > > > 3. After the rebuttal and discussion with the authors, I have a slightly positive view of the paper. However, I also found that other reviewers arguing for acceptance may not read the manuscript as carefully or be as familiar with link prediction as I am. For example, Reviewer fyCt commented that Link-MoE is trained end-to-end, which is in fact a two-stage training. Also, no other reviewer asks questions about time complexity. Even though the "time complexity" question can be an unfair gatekeeping for some studies, it is critical for almost all sota LP methods (BUDDY, NCN, LPFormer) after SEAL. Therefore, I have to make the decision carefully to reflect the actual merit of the paper. I need some time to consider the score and discuss with AC about the situation. I hope the authors understand.

---

> > > > > > > ### Author Response · Authors · 2024-08-14
> > > > > > > **Response to Reviewer SDXC**
> > > > > > >
> > > > > > > Dear Reviewer SDXC,
> > > > > > >
> > > > > > > Thank you for your further reply. We appreciate your engagement and would like to provide additional clarification:
> > > > > > >
> > > > > > > 1. As you pointed out, MoE "achieve better efficiency by routing the prediction to different domain experts.", our Link-MoE benefits from such sparse gating strategy. We have demonstrated that Link-MoE with top-2 gating achieves comparable performance to using all experts. Therefore, we respectfully disagree  that the Link-MoE is very bad at inference time.
> > > > > > >
> > > > > > > 2. We have indeed discussed Link-MoE with sparse gating and presented experimental results in **Section 5.2 and Appendix D of our paper**. However, the sparse gating is a widely used technique in MoE.  We didn't consider the sparse gating as the major contribution of our paper, which is why we included the related results in the Appendix. Instead, the key contributions and novelty of Link-MoE lie in the thorough analysis of the complementary properties of GNN4LP models, the insight that different node pairs benefit from different experts and heuristics, the exploration of the relationship between heuristics and GNN4LP models, the novel gating design, and the resulting strong performance.
> > > > > > >
> > > > > > > We understand your concern about the importance of time complexity in link prediction. However, Link-MoE can effectively benefit from the sparse gating mechanism commonly used in traditional MoE models. Link-MoE introduces a novel approach to link prediction from a different perspective:  instead of designing new models from scratch, we first focus on understanding the behavior of existing models and then leveraging their strengths for better predictions. We believe this idea can be applied to other tasks and domains.
> > > > > > >
> > > > > > > Best regards,
> > > > > > >
> > > > > > > All authors

---

> > > > > > > > ### Author Response · Authors · 2024-08-14
> > > > > > > >
> > > > > > > > Dear Reviewer SDXC,
> > > > > > > >
> > > > > > > > I believe that one goal of inviting multiple reviewers for one paper is to provide opinions from various perspectives and mitigate potential bias and unfairness because different people may view things from different angles.  We always believe the professional of reviewers that is why we are eager to submit our papers for pear reviews and feedback. So it is unfair for you to mention that "other reviewers arguing for acceptance may not read the manuscript as carefully or be as familiar with link prediction as I am."

---

### Official Review · Reviewer_758a · 2024-07-06

**Soundness:** 3
**Presentation:** 3
**Contribution:** 4
**Rating:** 7
**Confidence:** 4

**Summary:**

The paper proposes a mixture of experts model, Link-MoE, for link prediction on graphs. The motivation behind the proposed approach is that different node pairs within the same dataset necessitate varied pairwise information for accurate prediction, while existing methods consider the same pairwise information uniformly across all pairs. To address this, Link-MoE consists of multiple expert models (existing link prediction methods) and a gating model whose goal is to produce importance scores to weight the different expert models according to their contribution towards the final prediction. Experiments on several real-world datasets show that the proposed Link-MoE outperforms existing competing methods, validating the rationale behind its design.

**Strengths:**

S1) The source code is provided with the submission

S2) The paper is overall well-written

S3) The proposed approach is compared against several competing methods and baselines.

S3) Strong experimental results: the proposed method outperforms all the considered competing methods.

S4) The proposed approach is well motivated

S5) The proposed method was assessed on standard benchmark datasets for link prediction

**Weaknesses:**

W1) The figures are hardly readable sometimes (e.g., Fig. 8), the font size (in the legend) should be increased.

W2) Notation problem: in Eq. 1 the gating function G has only one parameter while in Eq. 2 G takes both x_ij and s_ij. Please uniform it as this may be a little confusing for the reader.

W3) The difference between the proposed Link-MoE and the considered baseline Global-Ensemble should be further discussed in the main paper given the strong similarity between the two approaches. As a suggestion, I recommend the authors to move lines 572-575 to the main paper since it describes the main differences between the two methods, i.e. the importance weight vector is uniform in Global-Ensemble while it is adaptable to the pair in Link-MoE.

W4) To evaluate the effectiveness of the proposed method different metrics are reported for different datasets.

Minor:
- Eq. 3: final extra parenthesis
- Line 220: “consits” -> “consists”
- Line 234: extra ”the” in “represent the all the”

**Questions:**

Q1) Why reporting different metrics for different datasets? As an example, line 155, the authors state that they report MRR for Citeseer and Hits@50 for ogbl-collab. Similarly, in Table 2, different metrics are reported for different datasets. have seen this approach used in other papers on link prediction, but I am not convinced that it is the correct way to proceed because, as shown in the tables in the appendix (e.g., Tables 5-7), the proposed method is not always the best when the evaluation metrics are changed.

**Limitations:**

Yes

---

> ### Author Rebuttal · Authors · 2024-08-07
>
> Dear Reviewer 758a,
>
> Thank you so much for your support and recognition of our framework. We are pleased to provide detailed responses to address your concerns.
>
> **W1: The figures are hardly readable sometimes (e.g., Fig. 8), the font size (in the legend) should be increased.**
>
> **A1:** Thank you for pointing out this issue. We will enlarge the font size of the axis labels, legend, and axis ticks for all figures to ensure they are more readable.
>
> **W2: Notation problem: in Eq. 1 the gating function G has only one parameter while in Eq. 2 G takes both $x_{ij}$ and $s_{ij}$. Please uniform it as this may be a little confusing for the reader.**
>
> **A2:** In Eq. 1, we initially used $h_{ij}$ to represent all the heuristics, including both the pairwise node features $x_{ij}$ and the structural heuristics $s_{ij}$.
> To alleviate potential confusion, we will rewrite Eq. 1 to match Eq. 2 as follows:
> $$
> Y\_{ij} = \sigma \left( \sum\_{o=1}^m G(\mathbf{x}\_{ij}, \mathbf{s}\_{ij})\_o E\_o(\mathbf{A}, \mathbf{X})\_{ij} \right)
> $$
>
> **W3: The difference between the proposed Link-MoE and the considered baseline Global-Ensemble should be further discussed in the main paper given the strong similarity between the two approaches.**
>
> **A3:** We will further discuss the Global-Ensemble method in the main text. Specifically, we will include a discussion of this method at the end of Section 3. We will first include the definition of the Global-Ensemble method (given by lines 570-575 in the appendix). We will then discuss the weaknesses of this method, mainly that the weights are non-adaptive and are the same for all links. Our empirical results in Section 3.1 and 3.2 also show that this kind of design is unable to model all links. The deficiency of this formulation serves as a strong motivation for our proposed method Link-MoE in Section 4.
>
> **W4: To evaluate the effectiveness of the proposed method different metrics are reported for different datasets.**
>
> **A4:** In Table 1, we use MRR for evaluating Cora, Citeseer, and Pubmed datasets, while Hits@50, Hits@100, and MRR are used for the ogbl-collab, ogbl-ppa, and ogbl-citation2 datasets. This approach is consistent with prior research, where notable methods [13, 14, 15, 16, 27, 53] have been designed for link prediction tasks. Additionally, we provide the results of other metrics in the Appendix, specifically in Tables 5, 6, 7, and 8.
>
> **Q1: Why reporting different metrics for different datasets?**
>
> **A5:** For the training of Link-MoE, we adhere to the experimental settings outlined in prior research [53], which serves as a benchmark for the link prediction task.
> During the training, we select and save the best model based on the validation performance. However, there are multiple evaluation metrics, which might be not aligned well. In this paper, we select the best model based on MRR for the Cora, Citeseer, and Pubmed datasets. For the ogbl-collab, ogbl-ppa, and ogbl-citation2 datasets, we used Hits@50, Hits@100, and MRR as selection metrics following [53].
> As a result, our method might not perform optimally on all individual metrics due to the misalignment of different metrics. Nevertheless, we consistently rank within the top 3 across almost all datasets and metrics. These observations indicate that Link-MoE is capable of delivering strong performance across various metrics.
>
> In the revision, we plan to fix all the typos and add another experiment that selects the best results based on each specific metric to provide a more comprehensive evaluation.
>
> We hope that we have addressed the concerns in your comments, and please kindly let us know if there is any further concern, and we are happy to clarify.

---

> > ### Comment · Reviewer_758a · 2024-08-08
> > **Follow-up discussion**
> >
> > I appreciate the detailed responses from the authors (as well as those to other reviewers), which have made me more enthusiastic about this work. I have two follow-up questions:
> > - I am curious about how you computed the Jaccard coefficient to calculate the overlap ratio between each pair of heuristics (Figures 2 and 3). Did you use a threshold to decide whether a prediction for an edge is ‘present’ or ‘missing’? Otherwise, I cannot see how you counted the number of edges that were correctly predicted or not by a pair of methods.
> > - Another point of curiosity: In line 129, you state that you assess the combination of a pair of heuristics by simply adding their original values. Does this mean that you did not normalize the scores before adding them? Since some heuristics may have different ranges, this approach might not be appropriate.

---

> > > ### Author Response · Authors · 2024-08-08
> > > **Response to the follow-up questions**
> > >
> > > Dear Reviewer 758a,
> > >
> > > Thank you for your thoughtful responses and your enthusiasm about our work. We are glad to answer your follow-up questions.
> > >
> > > **1. Computation of Jaccard Coefficient**
> > >
> > > For the calculation of the Jaccard coefficient, we use the Hits@K metric for each edge. Specifically, we choose Hits@3 for small datasets and Hits@20 for the OGB datasets. We first rank the prediction scores of each method for both positive and negative edges. If the prediction score of a positive edge is in the top-K, we label this positive edge as 'present' and add it to the correct prediction set. In this way, we can calculate the Jaccard coefficient by comparing the correct prediction sets for each pair of methods.
> > >
> > > Besides, your suggestion that using a threshold is also a feasible way.
> > >
> > > **2. Combination of Heuristics**
> > >
> > > Regarding the combination of heuristic scores, we did perform the normalization as you suggested. Specifically, we will normalize each Heuristic (H) to the range of [0, 1] using $\frac{H-Min\\_H}{Max\\_H - Min\\_H}$, where $Max\\_H$ and $Min\\_H$ are the maximum and minimum heuristic values in the dataset. One exception is the Shortest Path (SP), where a smaller SP indicates a higher likelihood that two nodes are connected. Therefore, we first calculate $\frac{1}{SP}$, and then normalize $\frac{1}{SP}$ in the same way as other Heuristics.
> > >
> > > We will add these details in our revision. Thank you once again for your thoughtful feedback and for helping us improve our work. If you have any further comments or questions, please let us know.
> > >
> > > Best regards,
> > >
> > > All Authors

---

### Official Review · Reviewer_fyCt · 2024-07-07

**Soundness:** 4
**Presentation:** 4
**Contribution:** 4
**Rating:** 8
**Confidence:** 4

**Summary:**

Link prediction in graphs is a fundamental task in graph machine learning and multiple heuristics and ML algorithms have been designed in research to leverage the pairwise and structural information to predict links between nodes. This work takes inspiration from the success of MoE models across various verticals and proposes a new method that outperforms the SoTA baselines on this task on several standard datasets. The authors demonstrate that heuristics and algorithms for LP are very diverse in their capabilities to predict links depending on the node structure and pairwise features (i.e. when one would perform well, the other might not). Due to this little overlap in their abilities the authors propose a dynamically weighted ensemble (MoE) like approach - such that a gating network predicts the weight given to each expert's prediction and the overall value is the weighted sum.
They also provide a very practical method to train the experts and the gating network achieving SoTA performance beating the second best by significant margins.

**Strengths:**

The paper is very well written with the motivations being made clear and experiments being clearly able to prove what the authors set out to. The number of experiments and metrics being covered are large enough to justify the claims in a border sense - with all the obvious baselines being addressed. I particularly liked the comparisons with the Mean and Global weight ensembles, and LPFormers.
The authors also provide an efficient and effective training strategy to build this E2E - which in my opinion can be understood and used easily by the community.
The paper also provides reasonable explanations for the results and conducts ablations (in section 5) required to improve understanding and readability for a potential user.
This result greatly improves on the SoTA baselines and should be generalisable to multiple graph use cases.

**Weaknesses:**

- The method being used is fairly obvious and intuitive.
- It solves for a very specific task - Link Prediction in an inductive setting only. Not clear to me how it could be leveraged for a transductive setting as well

**Questions:**

- Did you face any convergence issues with training the gating network ?
- What happens when you use a graph network which is transductive in nature ? Can that information be leveraged in some manner ?
- In the situations when the graph is a KG on language. Can LLM's outperform this method ?

**Limitations:**

The method is pretty generic and the societal impacts are not discussed in a lot of detail as it wont be different from existing methods.

---

> ### Author Rebuttal · Authors · 2024-08-07
>
> Dear Reviewer fyCt,
>
> We sincerely appreciate your recognition of our framework and insightful comments. We are pleased to provide detailed responses to your questions.
>
> **W2: It solves for a very specific task - Link Prediction in an inductive setting only. Not clear to me how it could be leveraged for a transductive setting as well.**
>
> **A1:** We'd like to clarify that our paper actually focuses on the transductive setting, where the same graph is used during both the training and testing phases. In contrast, the inductive setting involves unseen nodes and connections in the test graphs.  Since the vast majority of methods for link prediction tasks are designed specifically for the transductive setting (please refer to  [14, 15, 25, 26, 13, 16, 27] in our paper), we have limited our focus to it as well.
>
> However, due to the generality of our Link-MoE method, it should have no issue adapting to the inductive setting when the individual expert models are themselves inductive.  In the inductive setting, we would train the gating model based on the heuristics calculated from the training graph. During inference, we would simply recalculate the heuristics based on the test graph before applying Link-MoE. We believe this approach would allow Link-MoE to perform effectively in inductive settings, and we intend to explore this in future work.
>
> **Q1: Did you face any convergence issues with training the gating network?**
>
> **A2:** We did not encounter any convergence issues with training the gating network in our Link-MoE model. Our model successfully converges on the datasets we used, as evidenced by the smooth training loss curves on Pubmed presented in Figure 2 in the global response.
>
> **Q2: What happens when you use a graph network which is transductive in nature? Can that information be leveraged in some manner?**
>
> **A3:** As discussed in A1, while our Link-MoE focuses on the transductive setting, it can be generalized to the inductive setting. During the training phase, we train the gating model based on the heuristics calculated from the training graph. During inference, we would recalculate the heuristics based on the test graph before applying Link-MoE.
>
> **Q3: In the situations when the graph is a KG on language. Can LLM's outperform this method?**
>
> **A4:** We appreciate the inspiring question.
>
> Due to the uniqueness of KGs, the methods used for KGs differ from those used on non-KGs (i.e., the graphs used in our paper). This is because the type of structural information needed to accurately predict new links depends on the graph. On KGs, research shows that path-based information that connects the two nodes are necessary for strong performance [16]. On the other hand, for link prediction on non-KGs, heuristics such as common neighbors or feature similarity tend to be more important [17]. **Because of the disparity in inductive bias, in our paper we limit our focus to link prediction on non-KGs only**. However, due to the generality of our method Link-MoE, it should have no issue adapting to the link prediction on KGs, as we only need to use KG-specific methods as the experts.
>
> Lastly, as of now there is no consensus on whether LLMs can outperform graph-based methods for link prediction on KGs. However, recent work has shown that for inductive link prediction, LLM-based methods can potentially outperform SOTA graph-based methods on KGs (see [1] below).
>
> [1] Wang, Kai, et al. "LLM as Prompter: Low-resource Inductive Reasoning on Arbitrary Knowledge Graphs." arXiv preprint arXiv:2402.11804 (2024).

---

### Official Review · Reviewer_bwf6 · 2024-07-08

**Soundness:** 3
**Presentation:** 3
**Contribution:** 3
**Rating:** 5
**Confidence:** 5

**Summary:**

This paper presents a mixture of experts model, termed Link-MoE, for link prediction on graphs. Link-MoE individually trains various link prediction models as experts and selects the most appropriate experts for different node pairs. The prediction results from the selected experts are then weighted to produce the final predication. Experiments and analyses validate the effectiveness of the proposed method.

**Strengths:**

1. This paper is generally well-written and easy to follow. Necessary analyses are conducted to well motivate the design of the proposed method.
2. The proposed method is presented as a general framework, so that various link prediction models (i.e., experts) can be combined via different weights generated by the gating function.
3. The experimental design can largely validate the efficacy of the proposed method.

**Weaknesses:**

1. This paper presents a straightforward application of the MoE model to the problem of link prediction. Despite its empirical effectiveness, the technical novelty appears limited. The methodological contributions could be enhanced by addressing specific problems associated with MoE models, e.g., the collapse problem, in the context of link prediction.
2. The experimental evaluation could provide more insights and deeper analyses on important issues, for example,
- How does Link-MoE determine the appropriate number of experts?
- On heterophilic graphs, how the experts selected by Link-MoE differ from those selected on homophiles graphs?
3. The complexity of the proposed method can be high. The nature of using MoE makes the proposed method difficult to scale up.

**Questions:**

See weaknesses above.

**Limitations:**

The authors have discussed the limitations in terms of the scalability.

---

> ### Author Rebuttal · Authors · 2024-08-07
>
> Dear Reviewer bwf6,
>
> We appreciate your constructive feedback. We are pleased to provide detailed responses to address your concerns.
>
> **W1:Despite its empirical effectiveness, the technical novelty appears limited.**
>
> **A1:** The novelty of Link-MoE. Our Link-MoE model stands out by intelligently leveraging the complementary strengths of different GNN4LP models, resulting in substantial performance gains. The core of our model's success lies in the following findings and innovations:
>
> *1. GNN4LP Models are Complementary:* Our preliminary studies show that the overlapping between different GNN4LP models is notably low, shown in Figure 3, 9 and 10 in our paper, indicating these models offer complementary insights for link prediction. Different node pairs may be best addressed by different models. The critical challenge, then, is determining the most effective way to assign these node pairs to the appropriate GNN4LP models.
>
> *2. Heuristic-Informed Gating Mechanism:* Our preliminary studies revealed that different GNN4LP models perform differently across different heuristic groups, shown in Figure 4, 11 and 12 in our paper. By designing a gating model that intelligently applies these heuristics, we facilitate the optimal assignment of node pairs to their most suitable predictors. This strategic use of heuristics marks a significant departure from traditional MoE applications, which use the same input features for gating and experts [1]. We compare the MoE with traditional gating with Link-MoE. The results are shown in Table 1 (Traditional MoE) in the global response. Traditional MoE only results in comparable performance to the best single experts, while our approach yields superior performance.
>
> [1] Towards Understanding Mixture of Experts in Deep Learning. NeurIPS'22
>
> **The Collapse problem.** In traditional MoE models, the collapse problem [37] can occur when a single expert is consistently selected, resulting in the under-utilization and inadequate learning of other experts. However, we employ a two-step training strategy for the Link-MoE. First, we train each expert separately. Then we train the gating model to leverage the strengths of each expert effectively. This approach ensures that each expert is well-trained before the gating model is introduced. We empirically observe that there does not exist collapse problem in our model. As illustrated in Figure 8 of this paper, different experts are activated to model different node pairs.
>
> We also tested the end2end training of the experts and gating models, as shown in Table 3 and Figure 1 in the global response. Our results show that the convergence speed of different experts varies significantly, which often leads to the model collapsing to a single expert. As a result, the performance of end2end training is not as good as the proposed Link-MoE. Despite this, it remains an interesting and challenging idea to explore the effective end2end training.
>
> **W2.1:How does Link-MoE determine the appropriate number of experts?**
>
> **A2:** There are mainly three types of pairwise structural information in link-prediction tasks: local structure proximity (LSP), global structure proximity (GSP), and feature proximity (FP). Different GNN4LP models leverages different heuristic information. Based on these, we classify these methods into three groups: LSP (NeoGNN, NCN, NCNC, n2v), GSP (Seal, Buddy, NbfNet, PEG), and FP (MLP, GCN).
>
> For the selection of experts, it is suggested to include methods that cover all groups due to the complexity of connection patterns across different datasets. To verify this, we conducted an experiment on the ogbl-collab dataset. When using all the models, the Hits@50 score is 71.32. However, if we remove all LSP experts, the Hits@50 score drops to 67.93. Interestingly, if we include only one LSP expert, the Hits@50 score can reach 71.25. This demonstrates the necessity of local structure information for the ogbl-collab dataset, but it also indicates that only one model may be sufficient to capture this information effectively.
>
> Furthermore, our experiments show that using just a few experts (3 or 4) can achieve similar performance levels to using all experts, as shown in A4.
>
> **W2.2:On heterophilic graphs, how the experts selected by Link-MoE differ from those selected on homophiles graphs?**
>
> **A3:** We analyzed the weights generated by the gating mechanism on two heterophilic graphs, Chameleon and Squirrel. The results are shown in Figure 3 in the global response. We observed that feature proximity-based experts, such as MLP and GCN, are rarely employed for both datasets. This is consistent with the characteristics of heterophilic graphs, where connect nodes tend to have dissimilar features. These findings demonstrate the effectiveness of our gating design, as it accurately selects the appropriate experts for different types of graphs.
>
> **W3:The complexity of the proposed method can be high.**
>
> **A4:** The proposed Link-MoE model employs a two-step training strategy, which first trains the single experts and then trains a lightweight gating model. The most time-consuming part is the training of the single experts. However, as discussed in A2, there is no need to use all available experts, which can effectively reduce complexity and enhance scalability. we conduct experiments on ogbl-collab and Pubmed by only using 3 or 4 experts (please refer to Different number of experts for the expert selection details in the global response). The results shown in Table 1 (3 Experts & 4 Experts).
>
> From the results, we can find that only 3 or 4 experts can achieve comparable performance with using all experts in the paper. Additionally, the two-step training strategy of our Link-MoE model introduces another layer of efficiency by allowing the integration of pre-trained models. In real-world scenarios where models are continually updated, this approach is highly efficient: when a new model arrives, we only need to train the gating model.

---

### Author Rebuttal · Authors · 2024-08-07

# Global Response

We thank the reviewers for the valuable comments and suggestions. In this global response, we are willing to provide information about tables and figures in the rebuttal pdf file.

**Table 1** presents the results of traditional MoE, a few experts, and experts used as input features for the ogbl-collab and Pubmed datasets. The evaluation metrics are Hits@50 and MRR for ogbl-collab and Pubmed, respectively.

1. Traditional MoE. It uses the same input features for the gating model and experts. Traditional MoE only results in comparable performance to the best single expert, while our approach yields superior performance.

2. Different number of experts. For the ogbl-collab, we use MLP, NCNC, BUDDY (3 experts) and Neo-GNN (4 experts); for Pubmed, we use NCN, SEAL, NCNC (3 experts) and MLP (4 experts). From the results, we can find that only 3 or 4 experts can achieve comparable performance with using all experts in the paper. Notably, we don't use the computationally intensive SEAL for the ogbl-collab datasets. Furthermore, the inclusion of the less effective MLP expert in the Pubmed dataset still results in performance improvement, highlighting the complementary nature of the experts and the effectiveness of the proposed method.

3. With Experts as Input. We concatenate the prediction results of experts with the heuristic features as the input feature of gating model. Results show that this additional input did not lead to improvement.

**Table 2** compares the results of ensemble methods with our model. The evaluation metrics are Hits@50 and MRR for ogbl-collab and Pubmed, respectively. Ensemble [1] and Ensemble [2] refer to two ensemble methods specifically designed for link prediction tasks. As shown, our model outperforms these ensemble methods.

[1] Stacking Models for Nearly Optimal Link Prediction in Complex Networks, PNAS'20

[2] An Ensemble Model for Link Prediction based on Graph Embedding, Decision Support Systems'22

**Figure 1** is derived from  results when training experts and gating in an end2end way on ogbl-collab. Figure 1(a) shows the overall and each expert’s performance. Figure 1(b) shows the gating weights of each expert.

**Figure 2** illustrates the convergence process of training for our Link-MoE model on the Pubmed dataset, demonstrating a smooth training loss curve.

**Table 3** shows the results of end2end training on the ogbl-collab and Cora datasets. The evaluation metrics are Hits@50 and MRR for ogbl-collab and Cora.

1. Cora: we employed eight models (MLP, node2vec, GCN, NeoGNN, NCN, NCNC, SEAL, and BUDDY) and used the gating model to select the top-3 experts for each node pair.

2. ogbl-collab: we used four models (MLP, GCN, NCN, and NCNC) and selected the top-2 experts.

**Figure 3** displays the weight distribution generated by our gating model for two heterophilic graphs:  Chameleon and Squirrel datasets. We can see that feature proximity-based experts, such as MLP and GCN, are rarely employed for both datasets, which is consistent with the characteristics of heterophilic graphs, where connect nodes tend to have dissimilar features.

**Table 4** presents the results of our Link-MoE model under the HeaRT setting [53] for the ogbl-collab and Pubmed datasets. It is evident that our model achieves better performance even under this challenging HeaRT setting. This highlights the effectiveness of our approach in leveraging the strengths of multiple experts to achieve superior performance.

---

### Decision · Program_Chairs · 2024-09-25

**Decision:**

Accept (poster)

**Comment:**

The paper proposes an ensemble method for link prediction, where the mixing coefficients are predicted *separately* for each pair of nodes based on a set of heuristic features.
The strengths of the paper are
1. A clear motivation, including detailed and convincing preliminary analyses (`bwf6`, `fyCt`, `758a`, `SDXC`).
2. Strong experimental results (`bwf6`, `fyCt`, `758a`, `SDXC`) and evaluation on standard benchmark datasets (`758a`).
3. The availability of the source code (`bwf6`) and the simplicity of rebuilding the system in the community (`fyCt`), as well as being an intuitive method (`fyCt` - this was mentioned as a weakness, but to me it is a strength).
4. Clear writing (`bwf6`, `fyCt`, `758a`, `SDXC`).

The weaknesses are
1. Its limited technical novelty (`bwf6`, `SDXC`)
2. Its limitation to a single use case, inductive connection prediction (`fyCt`)
3. Its computational complexity (`bwf6`, `SDXC`)
4. The lack of additional analyses, e.g. the number of experts selected or qualitative differences in homophilic vs. heterophilic graphs.
5. Some missing baselines, namely [0] (`SDXC`) - during the rebuttal additional results for [1, 2] were provided instead.
6. The lack of a clear distinction again mixture of experts.

There were discussions with most reviewers, some over several rounds, in which some issues were resolved while others remained controversial.

[0] Ghasemian, A., Hosseinmardi, H., Galstyan, A., Airoldi, E. M., & Clauset, A. (2020). Stacking models for nearly optimal link prediction in complex networks. Proceedings of the National Academy of Sciences.
[1] Stacking Models for Nearly Optimal Link Prediction in Complex Networks, PNAS'20
[2] An Ensemble Model for Link Prediction based on Graph Embedding, Decision Support Systems'22